# RESILIBENCH: EVALUATING AGENTIC WORKFLOW ADAPTATION IN STOCHASTIC ENVIRONMENTS

**Ruicheng Ao**[1,*] **, Zeping Min**[2,*] **, Tingyu Zhu**[3,*] **, Wotao Yin**[2] **, Xinshang Wang**[2]

[1]Institute for Data, Systems, and Society, Massachusetts Institute of Technology,
[2]Alibaba Group US, DAMO Academy, [3]IEOR, BAIR, UC Berkeley

## ABSTRACT

We introduce ResiliBench, a benchmark that evaluates LLM workflow execution under simulated realistic conditions of instruction quality variability and tool execution uncertainty. Unlike existing benchmarks that encounter these challenges incidentally, our work makes uncertainty the primary focus of systematic study. The benchmark incorporates three key aspects: (1) modeling of probabilistic tool behaviors through parameterized error models that simulate real-world API failure patterns, (2) provision of MDP-derived workflows that maximize expected success rates, and (3) systematic evaluation of model robustness through controlled perturbations of workflow instruction quality. Our construction pipeline generates 5,040 tasks from a tool library of 30 APIs. The evaluation conducted across widely used large language models under conditions of probabilistic tool failures and varying instruction quality reveals notable performance differences. Specifically, MDP-optimal workflow prompts achieve an average success rate of 62.1%, Chain-of-Thought prompts yield an average success rate of 50.8%, and flawed workflow prompts result in an average success rate of 54.3%. Our benchmark is available at `https://github.com/Archer222arc/ResiliBench`.

## 1 INTRODUCTION

Large language models (LLMs) have shown capabilities in reasoning, planning, and tool utilization across diverse domains (Brown et al., 2020; Achiam et al., 2023). These models have demonstrated proficiency in applications that combine natural language understanding with tool orchestration (Schick et al., 2023; Qin et al., 2023; Parisi et al., 2022). Recent advances in agent-based systems have further expanded the scope of tool-use applications (Wang et al., 2023; Xi et al., 2023; Sumers et al., 2023). Workflow execution represents one area where LLMs must navigate sequences of tool interactions while maintaining coherence and achieving specified objectives, with applications ranging from software development (Fried et al., 2023; Nijkamp et al., 2022) to scientific computing (Lewkowycz et al., 2022).

When LLMs are deployed for workflow execution in production environments, they encounter three primary categories of challenges: (1) **Tool reliability issues**, where APIs exhibit probabilistic failures including timeouts, service outages, validation errors, and resource limitations that require adaptive error recovery strategies; and the model lacks higher-level information about tool reliability, for example, an LLM may not know the specific cause of an API interruption. (2) **Instruction quality variations**, where users provide instructions that may be incomplete, ambiguous, contradictory, or contain logical inconsistencies due to limited domain knowledge or communication constraints; and (3) **Complexity challenges**, where processes involve intricate tool dependencies, dynamic multi-agent coordination, and other highly complex tasks, and the number of dialogue turns or token limits restricts the model's exploration behavior.

Recent research has produced various tool-use benchmarks that evaluate LLMs' capacity for API interaction, multi-step reasoning, and workflow execution (Zhuang et al., 2023; Qin et al., 2023; Li et al., 2023; Patil et al., 2023), with specialized benchmarks emerging for mathematics (Hendrycks et al., 2021; Cobbe et al., 2021), code generation (Chen et al., 2021; Austin et al., 2021), and multi-

---

*Equal contribution

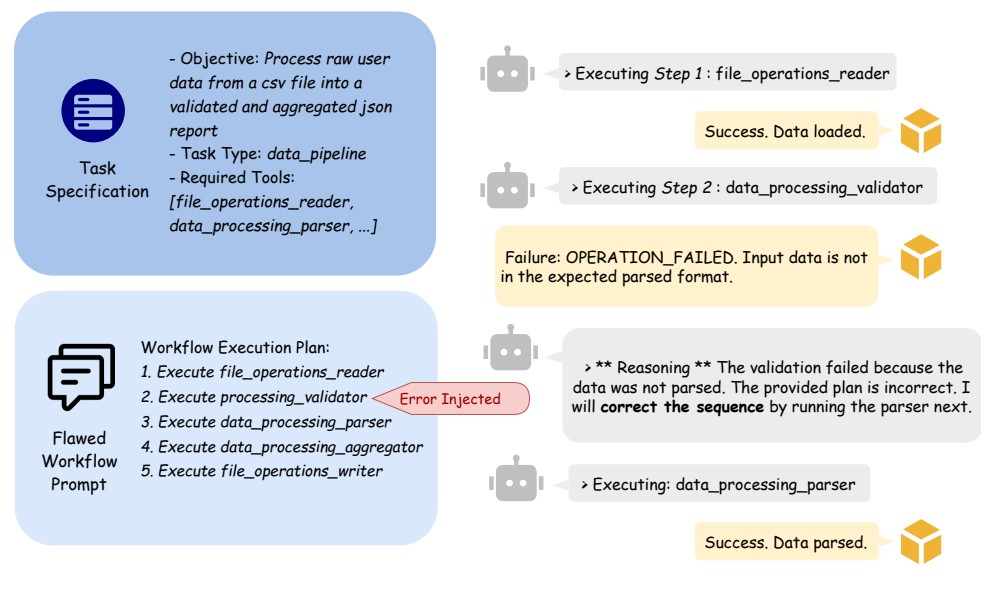

Figure 1: This figure demonstrates the ResiliBench setup. On the left, we provide task specifications, a tool library, and workflow prompts with injected errors. On the right, the LLM is expected to identify errors in the provided workflow, and execute with correction.

modal reasoning (Lu et al., 2022). These frameworks have made significant contributions to understanding LLM capabilities (Zhou et al., 2022; Wei et al., 2022), establishing important baselines for tool-use performance under controlled conditions.

We introduce ResiliBench*, a benchmark that systematically evaluates LLM workflow execution capabilities under realistic deployment conditions involving **tool reliability issues** and **instruction quality variations** (Liu et al., 2023; Jimenez et al., 2023). While existing benchmarks (Qin et al., 2023; Li et al., 2023; Huang et al., 2023) may encounter probabilistic tool failures and instruction variability as natural consequences of working with real-world APIs and diverse instruction generation, these works typically filter and curate APIs or instructions to minimize such issues. Our work makes two distinct contributions: Firstly, we make these uncertainties the primary focus of systematic study rather than attempting to filter them out. We deliberately introduce controlled perturbations through systematically generated flawed workflow instructions and probabilistic error models. This creates environments that require adaptive strategies, allowing us to measure the model's ability to recover from API errors and adaptively replan workflows to complete tasks.

Secondly, we provide analytically derived optimal workflows through MDP optimization. Under realistic deployment constraints, there exist three theoretical upper bounds for success rates: (i) 100%, which may not be achievable due to round/retry limits; (ii) the upper bound achieved by a policy knowing tool call procedures and able to anticipate tool errors (e.g., knowing the random seed of every tool operation); (iii) the upper bound achieved by a policy knowing tool call procedures but only aware of the probability of tool failure. Our MDP formulation provides a tractable approximation to bound (iii), computing workflows that maximize expected success rates by reasoning over known reliability statistics.

ResiliBench consists of three primary components: (1) task specifications spanning multiple types and complexity levels, (2) a tool registry modeling 30 APIs with probabilistic error behaviors and dependency constraints, and (3) multi-variant workflow guidance including both optimal execution

---

*ResiliBench was originally named PilotBench. The name was changed due to a naming conflict with another paper at ICLR 2026.

plans and perturbed variants. Each task is accompanied by four distinct prompt types—baseline, Chain-of-Thought, MDP-optimal workflow prompt, and flawed workflow prompt—designed to simulate low-quality prompts in real scenarios.

Our evaluation framework shows model families and size exhibit performance differences in current LLMs when confronted with realistic workflow environments. Assessments suggest performance changes when models encounter flawed instructions and probabilistic tool failures, with success rates that may correlate with both workflow instruction quality and tool reliability levels. These findings could inform approaches for improving LLM robustness in practical workflow execution scenarios.

The primary contributions of this work are multifold:

1. **Benchmark Design and Automated Construction**: We present ResiliBench, an evaluation framework that systematically assesses LLM workflow execution capabilities under instruction variability and tool uncertainty through automated task generation from structured tool libraries.

2. **MDP-Based Workflow Generation Framework**: We develop a Markov Decision Process (MDP) framework that generates theoretically optimal execution workflows maximizing expected success rates, along with seven types of systematically perturbed variants, enabling controlled evaluation of model robustness to instruction quality variations.

3. **Evaluation Findings**: Our evaluation reveals that models exhibit dramatically different robustness patterns when confronted with flawed instructions: GPT-4o-mini maintains relatively stable performance (optimal: 67.7%, flawed: 62.2%, a 5.5 percentage point drop), while Gemini-2.5-Flash shows substantial degradation (optimal: 60.1%, flawed: 20.0%, a 40.1 percentage point drop). We also observe emergent abilities in workflow execution where multi-step tool-use proficiency appears suddenly at certain parameter thresholds.

4. **Real-World API Integration**: We extend ResiliBench with a real-world task set that integrates live public APIs, serving as a complementary evaluation component that directly assesses model performance on actual API interactions. Real-world experiments reveal patterns consistent with the observations from the simulated API experiments.

The remainder of this paper is organized as follows. Section 2 details the core components, evaluation methodology, and benchmark statistics. Section 3 describes our automated construction pipeline and MDP-based workflow generation methods. Section 4 presents experimental results and analysis of current LLM capabilities. Section 5 reviews related benchmarks.

## 2 BENCHMARK SETUP

In this section, we present the setup of ResiliBench. We begin by describing the core components of the benchmark data and their organization, followed by the task types and evaluation methodology used to assess agent performance. A figure demonstrating the benchmark setup is provided in Figure 1.

### 2.1 BENCHMARK DATA ORGANIZATION

The ResiliBench dataset is structured around three primary components: task specifications, tool registry, and reference workflows.

**Task Specifications.** The benchmark comprises 5,040 unique tasks organized across multiple dimensions of type and complexity. Each task specification includes a natural language description, structured input/output requirements, a list of required tools, execution constraints, and comprehensive metadata. Detailed examples of task specifications are provided in Appendix B.1.

**Tool Registry.** The benchmark includes a comprehensive Tool Registry modeling 30 canonical software APIs with probabilistic behavior. Tools are systematically organized into six functional categories: data_processing, file_operations, network, computation, integration, and utility, with five tools per category ensuring balanced coverage.

Each tool definition specifies functional parameters with type constraints, structured return schemas, and explicit error models that enumerate possible failure modes. The system models five primary failure types: input validation failures (`INVALID_INPUT`), operational failures (`OPERATION_-FAILED`), timeout conditions (`TIMEOUT`), calculation errors (`CALCULATION_ERROR`), and resource overflow conditions (`OVERFLOW`). These error types reflect common failure patterns in real-world scenarios. Representative tool examples and detailed specifications are provided in Appendix B.1.

**Reference Workflows.** Each task is accompanied by four distinct prompt types (detailed in Section 3.3): baseline, Chain-of-Thought, MDP-optimal workflow prompt (optimal with respect to our MDP reward function), and flawed workflow prompt with systematic error injection. These variants enable evaluation of both instruction-following fidelity under high-quality instructions and robustness under flawed instruction quality.

## 2.2 TASK EXECUTION AND EVALUATION

**Execution Environment.** ResiliBench employs a simulated execution environment that provides realistic tool behavior. When agents invoke tools, a probabilistic simulator calculates success rates based on tool dependencies, execution history, and failure patterns, then generates appropriate success or failure responses. The simulator implements a base success rate of 0.8. Complete implementation details are provided in Appendix B.3.

**Evaluation Methodology.** Task outcomes are categorized into three distinct levels of success. A task is considered `full_success` if all specified tools are executed correctly and in the proper sequence. `Partial_success` reflects substantial task completion that satisfies most requirements but does not achieve perfect execution. `Failure` indicates that task completion is insufficient due to critical tool failures or breakdowns in execution.

The evaluation framework relies on four primary assessment criteria. *Required Tools Coverage* measures the proportion of tools successfully executed from the task's required tool list, with `full_success` requiring 100% coverage. *Sequence Correctness* assesses whether tools are executed in the exact order dictated by task dependencies. *Output Generation* verifies the successful execution of output-producing tools (e.g., writers, exporters, savers) as evidence of meaningful task completion. *Explicit Completion Signals* examines the conversation history for indications from the LLM that the task has been completed. Achieving `full_success` requires success in all four of these criteria.

Additionally, two considerations are also applied. *Minimum Tool Execution:* This defines task-type-specific thresholds for considering partial success. A task may be rated as `partial_success` if it meets at least two conditions, such as exceeding minimum tool execution coverage while producing the expected outputs. *Termination Conditions:* The evaluation also accounts for premature task termination, such as when a task experiences an excessive number of consecutive tool failures or becomes trapped in a repetitive loop, which typically results in a `failure` rating. Complete implementation details are provided in Appendix B.4.

## 2.3 BENCHMARK STATISTICS

ResiliBench contains 5,040 unique tasks systematically distributed across multiple dimensions to ensure balanced coverage of workflow scenarios. The benchmark incorporates 30 canonical tools organized into six functional categories, providing a controlled yet diverse environment for evaluating LLM workflow execution capabilities. Here we present some statistics about tasks and tools. For more detailed statistical information, please see Appendix B.1.

**Task Distribution.** We classify tasks using two approaches: task type classification and complexity classification. Task types include five categories: complex validation pipeline tasks, complex network integration tasks, basic file processing tasks, advanced computation pipelines, and simple data transformation tasks. Complexity classification spans three levels: easy, medium, and hard. Complex validation pipeline tasks (1,520 instances, 30.2%) form the largest category, followed by complex network integration tasks (1,360 instances, 27.0%), basic file processing tasks (1,200 instances, 23.8%), advanced computation pipelines (640 instances, 12.7%), and simple data transformation tasks (320 instances, 6.3%). Tasks span three complexity levels: easy (1,520 instances,

30.2%), medium (2,880 instances, 57.1%), and hard (640 instances, 12.7%). The distributions for both classification approaches are shown in Figure 2a and 2b.

**Tool Library Composition.** We organize tools into six categories: computation, data processing, file operations, integration, network, and utility, with each category containing five tools, as shown in Figure 2c. We also conduct a statistical analysis of tool-required parameters, with results presented in Figure 2d. The most common required parameter is `options`, which is required by 25 tools.

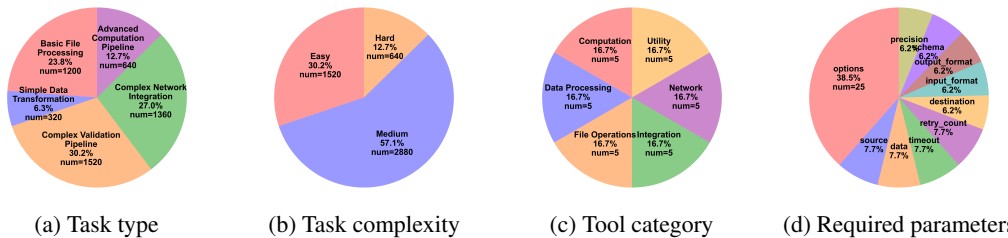

|        (a) Task type        |      (b) Task complexity      |       (c) Tool category       |   (d) Required parameters   |

Figure 2: ResiliBench statistics showing (a) task type distribution, (b) task complexity distribution, (c) tool category distribution, and (d) required parameter distribution.

## 3 BENCHMARK CONSTRUCTION

ResiliBench employs an automated construction pipeline that generates diverse workflow tasks from a systematic tool library. The construction process consists of three main stages: tool library generation, task creation, and prompt generation. This automated approach enables scalable benchmark expansion while maintaining task quality and diversity.

### 3.1 TOOL LIBRARY GENERATION

The tool library is constructed using a two-layer categorization system that balances systematic coverage with realistic workflow dependencies. The foundation consists of a category-operation matrix spanning 6 functional categories (data_processing, file_operations, network, computation, integration, utility) and 5 operations each, yielding 30 distinct tool types. Each tool follows the `{category}_{operation}` naming convention and includes standardized parameter templates, return schemas, and error handling specifications.

The second layer introduces semantic operation types that group tools by their workflow roles: sources (readers, fetchers), processors (parsers, transformers, analyzers), aggregators, outputs (writers, posters), and utilities. This semantic grouping enables dependency modeling where processors depend on sources, aggregators depend on processors, and outputs depend on aggregators. More detailed information can be found in Appendix C.1.

### 3.2 TASK GENERATION METHODOLOGY

Tasks are automatically generated through semantic matching between predefined operation sequences and the tool library. The system defines five task types: basic_file_processin, simple_data_transformation, complex_validation_pipeline, complex_network_integration, and advanced_computation_pipeline. Each task type follows a standard operation sequence, such as ['read', 'validate', 'transform', 'aggregate', 'write'] for complex validation pipelines.

The generation process uses RAG-based semantic matching to map operation steps to appropriate tools. For example, a 'read' operation can match to `file_operations_reader` or `network_fetcher` based on semantic similarity. Task variation is introduced through multiple tool choices per operation and different templates. An optional LLM enhancement phase improves task descriptions while preserving logical consistency, ensuring both natural language quality and structural integrity. More detailed information can be found in Appendix C.2.

Table 1: Comparison of prompt types in ResiliBench.

| Prompt Type | Base Components | Additional Elements | Evaluation Focus |
|---|---|---|---|
| Baseline | Task description, Input/output specs, Tool instructions | None | Basic instruction following |
| Chain-of-Thought | Task description, Input/output specs, Tool instructions | Explicit reasoning instructions | Reasoning and planning capabilities |
| MDP-Optimal Workflow | Task description, Input/output specs, Tool instructions | Detailed execution plan | Workflow adherence |
| Flawed Workflow | Task description, Input/output specs, Tool instructions | Perturbed execution plan with systematic errors | Error detection and robustness |

### 3.3 WORKFLOW PROMPT GENERATION

Each task has four distinct prompt types to evaluate different aspects of agent capability. Table 1 provides a comprehensive comparison of these prompt variants. Baseline prompts contain essential task information including description, input/output specifications, and tool usage instructions. Chain-of-Thought (CoT) prompts enhance baseline prompts with explicit CoT instructions that encourage step-by-step analysis. MDP-Optimal workflow prompts incorporate detailed execution plans derived from MDP formulations that account for tool dependencies and success probabilities.

The MDP-optimal workflow prompts are generated using a MDP framework that formally defines optimality in terms of expected cumulative reward for tool sequence selection under uncertainty. Our MDP formulation uses a composite state representation capturing tool execution states, progress tracking, etc. The action space consists of structured tool invocations. The system implements a two-phase adaptive reward strategy: an initial coverage-focused phase prioritizing tool discovery and usage, followed by a sequence-optimized phase emphasizing execution order and efficiency. Policy optimization employs Proximal Policy Optimization (PPO) with Transformer-based neural networks, mixed-precision training, and curriculum learning across five difficulty stages. The trained policy generates tool sequences that are optimal with respect to the learned reward function and state transition probabilities, producing workflows with maximized expected success rates given the MDP's modeling assumptions. More details can be found in Appendix D. Flawed workflow prompts are systematically generated by introducing controlled perturbations to MDP-optimal workflows across seven categories: sequential ordering errors, tool misuse, parameter configuration errors, missing critical steps, redundant operations, logic discontinuity, and semantic drift. More detailed information can be found in Appendix C.3.

### 3.4 REAL-WORLD API INTEGRATION

To complement our simulated environment, we extend ResiliBench with a real-world task set integrating live public APIs that directly assess practical deployment capabilities.

**API Selection and Characterization.** We source candidate APIs from the `public-apis` GitHub repository[†]. To identify APIs exhibiting probabilistic behaviors aligned with our benchmark design, we conduct empirical reliability assessment: each candidate API is invoked 20 times to measure success rate and response latency distributions. APIs are selected based on two criteria: (1) *Success rate variability*—non-deterministic behavior with occasional failures (timeouts, rate limiting, service interruptions), and (2) *Latency variability*—substantial variation in response times reflecting real-world network dynamics. This yields APIs naturally exhibiting stochastic behaviors that mirror our simulated error modes.

---

[†]`https://github.com/public-apis/public-apis`

**Task Construction.** We design 8 sequential workflow tasks based on 23 selected APIs' functionalities and output characteristics. Tasks are constructed by analyzing each API's input requirements, output schemas, and semantic capabilities to create realistic multi-step workflows. As an example, the `content_creation_task` requires LLMs to sequentially call four real-world APIs—fetching a random fact, a joke, a programming quote, and a stoic quote—then compile them into a social media post draft.

**Framework Integration.** We align real-world components with our existing infrastructure through: (1) constructing MCP-compliant tool registrations matching our simulated tool library schema (Section 3.1), including standardized parameters, return schemas, and error classifications, and (2) formatting tasks to match our task specification structure (Section 3.2). This enables uniform workflow prompt generation and evaluation methodology across simulated and real-world components. The specific tools and tasks are updated to our benchmark repository[‡].

## 4 EXPERIMENTS

### 4.1 EXPERIMENTAL SETUP

We evaluate several models spanning proprietary LLMs (GPT-4o-mini, O3-0416-Global, etc.) and open-source LLMs (DeepSeek-V3, Qwen2.5-32B, etc.). Each model is tested with 4 prompt variants per task (baseline prompt, Chain-of-Thought prompt, MDP-optimal workflow prompt, and flawed workflow prompt). We use a base success rate $\rho_{\text{base}} = 0.8$ for probabilistic tool execution.

The framework employs an interactive multi-turn execution environment with up to 10 conversational turns per task. Key implementation details include: (1) **Interaction protocol and execution**: Models use `<tool_search>query</tool_search>`, `<tool_info>tool_name</tool_info>`, and `<tool_call>tool_name</tool_call>` syntax; the system enforces one-tool-per-turn execution with dependency management. (2) **Automated feedback and parameter handling**: The system generates feedback messages with execution results, error information, and progress indicators; format detection reminders are provided when responses lack proper syntax. (3) **Error simulation**: Tool failures follow predefined error templates specific to each tool's MCP specification, including modes such as `TIMEOUT`, `DEPENDENCY_ERROR`, and `INVALID_INPUT`. For details, see Appendix B. In addition, we provide testing interaction examples in Appendix A.

Our experimental analysis of LLM workflow execution capabilities demonstrates:

- **Different robustness patterns to instruction quality**: Models exhibit dramatically different robustness patterns when confronted with flawed instructions. For example, GPT-4o-mini maintains relatively stable performance across instruction quality variations (optimal: 67.7%, flawed: 62.2%, a 5.5 percentage point drop), while Gemini-2.5-Flash shows substantial degradation (optimal: 60.1%, flawed: 20.0%, a 40.1 percentage point drop) (Section 4.3).

- **Emergent abilities of workflow execution**: Through experiments on the Qwen2.5 series, we observe emergent abilities in workflow execution, where the model's multi-step workflow execution ability appears suddenly at certain parameter thresholds rather than scaling smoothly (Section 4.4).

- **Real-world validation**: We extend our evaluation with a real-world test set integrating live public APIs. Real-world experiments reveal patterns consistent with the observations from the simulated API experiments (Section 4.5 and Appendix E.3).

### 4.2 OVERALL PERFORMANCE RESULTS

Table 2 presents the performance results using baseline, Chain-of-Thought, and MDP-optimal workflow prompts. When the models are presented with the MDP-optimal prompt, success rates range from 56.8% to 67.7% across both proprietary and open-source models. Among proprietary models, GPT-4o-mini achieves a 67.7% success rate, followed by GPT-5-mini (60.7%) and Gemini-2.5-Flash (60.1%). Open-source models show competitive performance, with Llama-3.3-70B reaching 66.1%.

---

[‡] https://github.com/Archer222arc/ResiliBench

Table 2: Performance: Baseline, Chain-of-Thought, and MDP-Optimal Workflow prompting.

| Model | Baseline | | | Chain-of-Thought | | | Optimal Workflow | | |
|---|---|---|---|---|---|---|---|---|---|
| | Full | Partial | Fail | Full | Partial | Fail | Full | Partial | Fail |
| GPT-4o-mini | 50.5 | 46.5 | 3.0 | 56.1 | 43.9 | 0.0 | 67.7 | 31.2 | 1.1 |
| O3-0416-Global | 52.7 | 44.9 | 2.4 | 48.9 | 45.6 | 5.6 | 58.5 | 35.1 | 6.4 |
| Gemini-2.5-Flash | 54.3 | 44.5 | 1.1 | 51.1 | 44.7 | 4.2 | 60.1 | 36.7 | 3.3 |
| GPT-5-mini | 52.0 | 46.0 | 2.0 | 54.3 | 45.7 | 0.0 | 60.7 | 35.5 | 3.8 |
| Llama-3.3-70B | 47.8 | 42.5 | 9.6 | 43.6 | 43.6 | 12.7 | 66.1 | 30.9 | 3.0 |
| Qwen2.5-32B | 52.5 | 43.8 | 3.7 | 51.7 | 45.0 | 3.3 | 65.0 | 31.9 | 3.1 |
| DeepSeek-V3 | 50.0 | 50.0 | 0.0 | 50.0 | 46.9 | 3.1 | 56.8 | 39.0 | 4.2 |
| Avg | 51.4 | 45.5 | 3.1 | 50.8 | 45.1 | 4.1 | **62.1** | 34.3 | 3.6 |

## 4.3 SENSITIVITY ANALYSIS

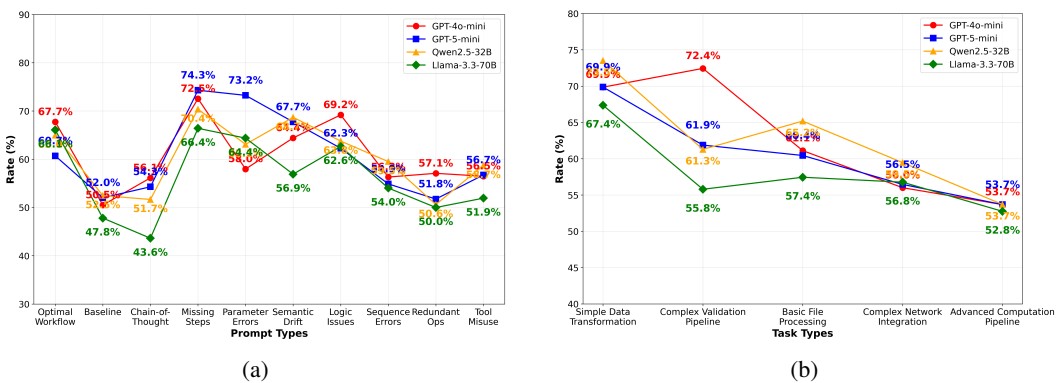

Figure 3: Performance analysis across representative models: (a) Prompt robustness analysis across 7 categories of flawed workflow prompts, and (b) model performance across different task types.

We conduct sensitivity analysis to examine model performance across two dimensions: instruction quality variation and task complexity progression. This analysis evaluates performance under the MDP-optimal workflow and flawed workflow prompts, and five task complexity levels.

**Instructions Quality Sensitivity.** Table 3 shows that MDP-optimal workflow prompts achieves higher average success rate (62.1%) compared to flawed workflow prompts (54.3%). Besides, models exhibit dramatically different robustness patterns when confronted with flawed instructions. Specifically, GPT-4o-mini maintains relatively stable performance with only a 5.5 percentage point drop (optimal: 67.7%, flawed: 62.2%), suggesting implicit error correction capabilities. In contrast, Gemini-2.5-Flash shows substantial performance degradation with a 40.1 percentage point drop (optimal: 60.1%, flawed: 20.0%). This indicates that tolerance to flawed instructions represents a distinct capability dimension. To analyze the specific failure modes, we examine seven types of systematic perturbations in Figure 3a, revealing that advanced models are particularly resilient to ordering and parameter errors but more vulnerable to semantic drift.

**Task Complexity Sensitivity.** Figure 3b demonstrates consistent performance degradation patterns as task complexity increases. Representative models show performance decline from simple content analysis tasks to complex computation pipelines: GPT-4o-mini (72.4% to 53.7%), GPT-5-mini (69.9% to 53.7%), Qwen2.5-32B (73.5% to 53.7%), and Llama-3.3-70B (67.4% to 52.8%). This degradation reflects the increasing cognitive demands of workflow execution, where advanced computation pipelines require more sophisticated reasoning about tool dependencies and execution chains compared to basic file processing tasks.

## 4.4 SCALING ANALYSIS

We conducted additional experiments on the Qwen2.5 series to investigate the relationship between model size and workflow execution capabilities. The Qwen2.5 series exhibits non-linear scaling patterns (Table 4), with performance varying substantially: 0.5% (3B), 63.5% (7B), 65.0% (32B), and 65.0% (72B).

The substantial jump from 3B to 7B (63.0 percentage point increase) suggests the emergence of basic tool comprehension capabilities around this scale, while performance plateaus from 32B to 72B indicate diminishing returns. This pattern reflects emergent abilities in workflow execution, where tool-use proficiency appears suddenly at certain parameter thresholds rather than scaling smoothly. This observation aligns with discussions such as (Berti et al., 2025).

Table 3: Performance comparison: Optimal vs. Flawed Workflow prompting.

| Model | Optimal Workflow | | | Flawed Workflow | | |
|---|---|---|---|---|---|---|
| | Full | Partial | Fail | Full | Partial | Fail |
| **GPT-4o-mini** | 67.7 | 31.2 | 1.1 | 62.2 | 34.6 | 3.2 |
| **O3-0416-Global** | 58.5 | 35.1 | 6.4 | 53.8 | 39.2 | 7.0 |
| **Gemini-2.5-Flash** | 60.1 | 36.7 | 3.3 | 20.0 | 12.8 | 67.3 |
| **GPT-5-mini** | 60.7 | 35.5 | 3.8 | 63.5 | 36.1 | 0.4 |
| **Llama-3.3-70B** | 66.1 | 30.9 | 3.0 | 59.5 | 36.2 | 4.4 |
| **Qwen2.5-32B** | 65.0 | 31.9 | 3.1 | 62.9 | 35.9 | 1.2 |
| **DeepSeek-V3** | 56.8 | 39.0 | 4.2 | 58.4 | 39.7 | 1.9 |
| **Avg** | 62.1 | 34.3 | 3.6 | 54.3 | 33.5 | 12.2 |

Table 4: Qwen2.5 series scaling analysis with detailed performance metrics.

| Model Size | Full Success Rate | Partial Success Rate | Failure Rate |
|---|---|---|---|
| Qwen2.5-3B | 0.5% | 0.5% | 99.1% |
| Qwen2.5-7B | 63.5% | 30.2% | 6.3% |
| Qwen2.5-32B | 65.0% | 31.9% | 3.1% |
| Qwen2.5-72B | 65.0% | 32.0% | 3.0% |

## 4.5 RESULTS ON REAL-WORLD TASK SET

To assess the transferability of our simulation findings, we also evaluate the models on real-world tasks. Table 5 presents performance across 8 tasks using 23 live public APIs. Consistent with our simulation results, models exhibit different robustness patterns when confronted with flawed instructions. Specifically, GPT-4o-mini shows moderate degradation (optimal: 42.1%, flawed: 34.3%, a 7.8 percentage point drop), while Gemini-2.5-Flash exhibits more substantial degradation (optimal: 55.3%, flawed: 34.1%, a 21.2 percentage point drop). We present the specific distribution of API error rates and types occurred in our experiments in Section E.4.

## 5 RELATED WORK

Recent studies on large language models (LLMs) have produced various benchmarks and frameworks for evaluating tool-use reasoning, workflow execution, and LLM robustness in complex tasks. For example, ToolQA (Zhuang et al., 2023) analyzes tool-use reasoning by differentiating between knowledge-based and tool-reliant questions but is limited to single tool invocations.

Some advanced benchmarks attempt to evaluate LLMs in more realistic scenarios. ToolBench (Qin et al., 2023) extends evaluations to a broader set of APIs and enable both single- and multi-tool tasks with curated APIs and instructions. MetaTool (Wang et al., 2024) uses meta-task augmentation to improve tool-use knowledge transfer. ToolAlpaca (Tang et al., 2023) studies generalization in

Table 5: Real-word Task Set Model Performance

| Model | Optimal Workflow | | | Flawed Workflow | | |
|---|---|---|---|---|---|---|
| | Full | Partial | Fail | Full | Partial | Fail |
| **GPT-4o-mini** | 42.1 | 38.6 | 19.3 | 34.3 | 34.3 | 31.4 |
| **Gemini-2.5-Flash** | 55.3 | 41.2 | 3.5 | 34.1 | 33.5 | 32.4 |
| **GPT-5-mini** | 47.6 | 45.2 | 7.1 | 36.6 | 36.3 | 27.0 |
| **Llama-3.3-70B** | 39.6 | 27.1 | 33.3 | 25.4 | 25.4 | 49.3 |
| **Qwen2.5-32B** | 55.3 | 44.7 | 0.0 | 42.3 | 38.8 | 18.9 |
| **DeepSeek-V3** | 40.2 | 39.3 | 20.5 | 36.0 | 35.4 | 28.6 |
| **Avg** | 46.7 | 39.4 | 13.9 | 34.8 | 34.0 | 31.3 |

Table 6: Comparison of workflow-oriented LLM benchmarks. Comparison involves 5 aspects: **Planning**: benchmark requires the LLM to design a sequence of tool calls. **Tool Choice**: benchmark measures correct selection of the appropriate tool/API. **Tool Call**: benchmark checks syntactic and semantic correctness of each invocation. **Multi-Step**: tasks demand two or more consecutive calls. **MCP Protocol**: tools are specified with the Model Context Protocol (or an equivalent structured schema).

| Benchmark | Planning | Tool Choice | Tool Call | Multi-Step | MCP Protocol |
|---|---|---|---|---|---|
| ToolQA(Zhuang et al., 2023) | No | Yes | No | No | No |
| ToolBench(Qin et al., 2023) | Yes | Yes | Yes | Yes | No |
| MetaTool(Wang et al., 2024) | No | Yes | No | No | No |
| ToolAlpaca(Tang et al., 2023) | No | No | Yes | No | No |
| API-Bank(Li et al., 2023) | Yes | Yes | Yes | Yes | Yes |
| SOP-Bench (Nandi et al., 2025) | Yes | Yes | Yes | Yes | No |
| Multi-Mission Tool Bench (Yu et al., 2025) | Yes | Yes | Yes | Yes | No |
| TheAgentCompany (Xu et al., 2024) | Yes | Yes | No | Yes | No |
| **Ours (this work)** | Yes | Yes | Yes | Yes | Yes |

smaller models, mainly focusing on simple tool use. Broader benchmarks such as API-Bank (Li et al., 2023), SOP-Bench (Nandi et al., 2025), and Multi-Mission Tool Bench (Yu et al., 2025) examine planning, procedural adherence, and adaptability, yet mostly emphasize dialog, industrial SOPs, or changing missions. TheAgentCompany (Xu et al., 2024) studies real-world digital tasks but does not deeply examine workflow quality and its effect on decision-making. We present a detailed comparison between our benchmark and others in Table 6.

## 6 CONCLUSION

We introduce ResiliBench, a benchmark with 5,040 tasks from 30 APIs that systematically evaluates LLM workflow execution under tool reliability issues and instruction quality variations. Our evaluation reveals dramatically different robustness patterns across model families and emergent abilities where multi-step proficiency appears suddenly at certain parameter thresholds.

## REPRODUCIBILITY STATEMENT

To ensure the reproducibility of our work, all components of the ResiliBench benchmark are publicly available. The complete tool library, the full set of 5,040 generated task specifications, and the source code for our automated construction pipeline are accessible at our anonymous GitHub repository: `https://github.com/Archer222arc/ResiliBench`. Furthermore, we have provided extensive details of our methodology in the appendices.

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

## A EXECUTION EXAMPLES

To provide concrete insight into the testing process, we present two actual execution examples from our evaluation logs for the gpt-4o model with our optimal prompt. These examples illustrate both successful and challenging scenarios that models encounter during advanced computation pipeline tasks.

### A.1 SUCCESSFUL EXECUTION EXAMPLE

In the following example, we provide an MDP-optimal workflow prompt to the LLM. The basic file processing task documentary is presented as follows.

```
Task Type: content_analysis
Prompt Type: optimal

Task Instance:
----------------------------------------
Required Tools: ['file_operations_reader',
'data_processing_parser', 'data_processing_filter']

Description: Load server log files (raw text/JSON) and recent
customer feedback forms from the database, parse and normalize
timestamps and service identifiers, and convert both sources into
a common CSV schema. Apply business rules to filter by production
services, map error codes to severity levels, and discard
debug-level noise. Aggregate response times, error counts, and
complaint keywords by hourly and daily intervals, compute rolling
averages and spike detection, and correlate spikes with customer
complaints. Deliver a trend analysis CSV and a one-page summary
highlighting recurring anomalies, peak degradation windows, and
suggested next steps for the IT ops team.
```

The prompt that we provide to the LLM consists of the description of the task, a detailed guide on how to perform tool search and tool call, and a recommended optimal sequence for executing the tools, given as

```
Workflow Execution Plan:

1. Execute file_operations_reader
2. Execute data_processing_parser
3. Execute data_processing_filter
```

The execution flow in this example begins with Turn 1,

```
Assistant: <tool_search>file operations reader</tool_search>
```

This tool search request from the agent triggers the search engine to provide a set of tool search results:

```
file_operations_reader, file_operations_writer,
file_operations_converter, file_operations_scanner,
file_operations_compressor
```

As an example, the full information of file_operations_reader returned by the tool search operation is presented as follows.

```
1. file_operations_reader
   Category: file_operations
```

```
Description: This tool is specifically designed to read and
retrieve data from files.
Relevance: 0.70
Parameters:
   - source (string): Source location or identifier [REQUIRED]
   - options (object): Additional options [OPTIONAL] default={}
Returns:
   - success (boolean): Whether operation succeeded
   - data (object): Retrieved or parsed data
   - metadata (object): Operation metadata
Possible Errors:
   - INVALID_INPUT: Input validation failed
   - OPERATION_FAILED: Operation could not be completed
   - TIMEOUT: Operation timed out
   - FILE_NOT_FOUND: Specified file not found
   - PERMISSION_DENIED: Insufficient permissions
```

The assistant then applies the tool call operation

```
<tool_call>file_operations_reader</tool_call>
```

followed by a success message indicating that the LLM should move to the next execution step:

```
User: file_operations_reader executed successfully.
Output: {
  "status": "completed",
  "tool": "file_operations_reader",
  "execution_context": {
    "step": 0,
    "total_tools_executed": 0
  }
}
```

```
Current progress: 1 tools executed. Continue with the next step.
```

After several turns of tool search and tool call, the final successful execution history consists of the following records:

```
Step 1: {'tool': 'file_operations_reader', 'success': True,
'output': "{'status': 'completed', 'tool':
'file_operations_reader', 'execution_context': {'step': 0,
'total_tools_executed': 0}}", 'error': None}

Step 2: {'tool': 'data_processing_parser', 'success': True,
'output': "{'status': 'completed', 'tool':
'data_processing_parser', 'execution_context': {'step': 1,
'total_tools_executed': 1}}", 'error': None}

Step 3: {'tool': 'data_processing_filter', 'success': True,
'output': "{'status': 'completed', 'tool':
'data_processing_filter', 'execution_context': {'step': 2,
'total_tools_executed': 2}}", 'error': None}
```

## A.2 CHALLENGING EXECUTION EXAMPLE

To illustrate common execution difficulties, we also present a challenging scenario where typical failure patterns occur.

The task instance is given as:

```
Task Instance:
----------------------------------------
Required Tools: ['network_fetcher', 'data_processing_validator',
'data_processing_parser']

Description: Pull active customer records from the CRM API and
subscriber lists from the email marketing platform, and retrieve
shipping-rate quotes for each customer address via the shipping
rate API. Normalize and convert JSON/CSV fields, deduplicate by
email, and map CRM fields to email platform fields (first_name,
last_name, email, subscription_status, lifetime_value,
shipping_class). Apply business rules: prefer the record with the
latest CRM last_updated timestamp, set shipping_class = "fast" if
lowest quote < $10, and mark lifetime_value > $200 as VIP. Push
updates to both systems via their APIs and produce a CSV of
synchronized records plus a small sync report with counts and
errors.
```

The reference workflow:

```
1. Execute network_fetcher
2. Execute data_processing_validator
     - Requires: data_processing_parser
3. Execute data_processing_parser
```

The execution results are provided as follows. For simplicity, we omit numerical details, keeping only the tool search and tool call decisions made by the model, and the execution result records.

```
Turn 1:
Assistant: <tool_search>network fetcher</tool_search>
User: Tool Search Results:  network_fetcher, network_poster,
network_monitor, network_router, utility_tracker (along with tool
information details)

Turn 2:
Assistant: <tool_call>network_fetcher</tool_call>
User: network_fetcher executed successfully.

Turn 3:
Assistant: <tool_search>data processing validator</tool_search>
User: Tool Search Results: data_processing_validator,
network_validator, data_processing_filter,
data_processing_aggregator, data_processing_parser

Turn 4:
Assistant: <tool_call>data_processing_validator</tool_call>
User: data_processing_validator failed due to dependency issues.

- DEPENDENCY PROBLEMS:
Missing dependency: 'data_processing_parser' must be executed
first

- RECOMMENDATION:
Execute the required dependencies first, then retry this tool.

Turn 5:
Assistant: <tool_search>data processing parser</tool_search>
User: Tool Search Results: data_processing_parser,
data_processing_aggregator, data_processing_transformer,
data_processing_filter, data_processing_validator
```

```
Turn 6:
Assistant: <tool_call>data_processing_parser</tool_call>
User: data_processing_parser failed.
Error: TIMEOUT: Operation timed out (after 26 seconds)

Turn 7:
Assistant: <tool_search>data processing aggregator</tool_search>
User: Tool Search Results: data_processing_aggregator,
data_processing_filter,
data_processing_parser, data_processing_transformer,
data_processing_validator

Turn 8:
Assistant: <tool_call>data_processing_aggregator</tool_call>
User: data_processing_aggregator failed due to dependency issues.

- DEPENDENCY PROBLEMS:
Dependency 'data_processing_parser' failed - cannot proceed safely

- RECOMMENDATION:
Execute the required dependencies first, then retry this tool.

Turn 9:
Assistant: <tool_call>data_processing_parser</tool_call>
User: data_processing_parser failed.
Error: INVALID_INPUT: Input validation failed (expected: JSON
format)

...
```

The agent's failure stemmed from two critical errors. First, in Turn 4, it overlooked the prerequisite that data_processing_parser must precede data_processing_validator, leading to an execution failure. Second, when data_processing_validator encountered a simulated timeout in Turn 6, the agent attempted an improper fallback. It invoked data_processing_aggregator, a tool for which it lacked the necessary information to correctly execute, causing another execution error. These cascading failures ultimately caused the agent to exhaust its allotted turns and fail the task.

## B  BASIC TASK FUNCTION COMPONENTS

The most basic component of the ResiliBench is a collection of *tasks*, presented as items in a task library. In the procedure of interacting with and evaluating an LLM, each task is mapped to multiple types of prompts, guiding the LLM to complete the task. The LLM is expected to perform correct tool calls, check the returned results, until all steps of the task are correctly finished.

In the remainder of this section, we present the task specifications, including their structure and the completion process for LLMs. We describe the tool library provided to LLMs and the simulated environments that facilitate tool execution and task completion. Finally, we explain how task execution results are generated and evaluated.

### B.1  TASKS AND TOOLS

#### B.1.1  PARAMETERS AND STATISTICS OF TASKS

This section provides a comprehensive overview of the task library, a collection of 5040 unique tasks designed for various purposes. The library is generated from multiple source files and encompasses a range of task types and complexities.

**Library Composition and Statistics.**    The tasks within the library are categorized by type, complexity. Table 7 provides a detailed breakdown of these distributions.

Table 7: Task library statistics.

| Category | Subcategory | Number of Tasks |
|---|---|---|
| **Task Type** | basic_file_processing | 1200 |
| | simple_data_transformation | 320 |
| | complex_validation_pipeline | 1520 |
| | complex_network_integration | 1360 |
| | advanced_computation_pipeline | 640 |
| **Complexity** | easy | 1520 |
| | medium | 2880 |
| | hard | 640 |

The tasks are generated based on several templates defined through the generation process. Each **Task Type** corresponds to a specific generation logic that determines the complexity and nature of the task.

**basic_file_processing:** These are fundamental tasks generated to represent simple data processing workflows. They typically involve a small number of tools (2-3) chosen randomly from the available tool library. The complexity is generally set to `easy`, focusing on straightforward input, processing, and output steps.

**simple_data_transformation:** This category includes tasks that use either one or two tools. Single-tool tasks are created for each tool category, often paired with optional validation or logging tools. Dual-tool tasks combine two randomly selected tools for a two-step process. These are also considered `easy` in complexity.

**complex_validation_pipeline:** These tasks simulate a multi-stage data validation pipeline. They are constructed by semantically selecting a chain of tools that follow a logical data flow: reading/parsing, transforming/validating, and writing/exporting. These tasks are assigned a `medium` complexity.

**complex_network_integration:** This task type is designed to simulate interactions with network APIs. The generator selects tools to create a sequence of fetching data from an endpoint, validating the response, and potentially posting data back. These are also of `medium` complexity.

**advanced_computation_pipeline:** These represent the most complex tasks. They are generated by identifying longer, more intricate chains of tools based on their dependencies. The generator attempts to find paths in the tool dependency graph, resulting in tasks that require a sequence of multiple tools to complete. These tasks are designated as `hard` complexity.

**Task Parameters.**    Each task in the library is defined by a set of parameters that describe its characteristics, requirements, and expected outcomes. Table 8 provides an overview of these parameters.

Table 8: Description of task parameters.

| Parameter | Type | Description |
|---|---|---|
| instance_id | String | A unique identifier for the task instance. |
| task_type | String | The category of the task (e.g., `basic_file_processing`, `complex_validation_pipeline`). |
| description | String | A brief, human-readable description of the task. |
| inputs | Object | An object containing the input data and options for the task. This often includes a `source` file path and various processing `options`. |
| expected_outputs | Object | An object describing the expected outcome of the task, which typically includes a `success` boolean and may contain `metadata` about the results. |

Table 8 continued from previous page.

| Parameter | Type | Description |
|---|---|---|
| required_tools | Array | A list of strings, where each string is the name of a tool required to complete the task. |
| constraints | Object | An object specifying any constraints on the task execution, such as `timeout` in seconds and the maximum number of `max_retries`. |
| complexity | String | The complexity level of the task, categorized as `easy`, `medium`, or `hard`. |
| metadata | Object | An object containing metadata about the task generation, such as the `template` used, generation timestamp, and whether it was LLM-generated. |
| original_description | String | The original, more detailed description of the task before any enhancements or modifications. |

### B.1.2 EXAMPLE OF A TASK

To directly provide an idea of what a task might look like, we present an example of a "basic file processing task" in the task library.

The following task requires the LLM to execute a sequence of `file_operations_reader`, `data_processing_parser` and `data_processing_filter` tools to process the input data.

```json
{
    "instance_id": "task_dee2d02d",
    "task_type": "basic_file_processing",
    "description": "This task retrieves data from a CSV file,
    parses it into a structured format, and filters it based on
    specified criteria. The final result will be a refined
    dataset that meets the filtering conditions.",
    "inputs": {
      "source": "data/input_file.csv",
      "options": {
        "filter": "true"
      }
    },
    "expected_outputs": {
      "success": true,
      "metadata": {
        "total_records": 100,
        "filtered_records": 80
      }
    },
    "required_tools": [
      "file_operations_reader",
      "data_processing_parser",
      "data_processing_filter"
    ],
    "constraints": {
      "timeout": 300,
      "max_retries": 3
    },
    "complexity": "easy",
    "metadata": {
      "template": "basic_file_processing",
      "generated_at": "2025-07-10T04:27:35.913857",
      "timeout": 300,
```

```
        "semantic_generation": true,
        "llm_generated": true,
        "inputs_generated_from": "llm"
    }
}
```

### B.1.3 PARAMETERS AND STATISTICS OF TOOLS

The tool library consists of a set of 30 canonical tools, each designed to perform a specific operation. These tools are categorized based on their functionality, such as data processing, file operations, and network interactions. This section provides a detailed breakdown of the library's composition and the parameters used across the different tools.

**Tool Library Composition.** The tools are grouped into six distinct categories. The distribution of tools across these categories is uniform, with each category containing five tools. This balanced distribution ensures a wide range of capabilities within the library. Table 9 summarizes this distribution.

Table 9: Distribution of tools by category.

| Category | Number of Tools |
|---|---|
| computation | 5 |
| data_processing | 5 |
| file_operations | 5 |
| integration | 5 |
| network | 5 |
| utility | 5 |

**Common Tool Parameters.** The tools in the library share a common set of parameters that define their inputs and behavior. These parameters allow for consistent interaction with the tools, regardless of their specific function. Table 10 provides a description of the most frequently used parameters.

Table 10: Common tool parameters and their usage.

| Parameter | Data Type | Occurrences | Description |
|---|---|---|---|
| options | object | 25 | A flexible object for passing additional options. |
| source | string | 5 | The source location or identifier for data input. |
| data | object | 5 | The data payload to be processed or sent by the tool. |
| timeout | number | 5 | The timeout duration for the operation, in seconds. |
| retry_count | number | 5 | The number of times to retry the operation upon failure. |
| destination | string | 4 | The target location or identifier for the output. |
| input_format | string | 4 | The format of the input data (e.g., JSON, CSV). |
| output_format | string | 4 | The desired format for the output data. |
| schema | object | 4 | The validation schema to check the data against. |
| precision | number | 4 | The numerical precision for computational tasks. |

**Tool Construction Parameter Details.** Each tool is defined by a structured set of parameters, return values, and potential errors. The following table, Table 11, provides a comprehensive overview of the fields that constitute a tool's definition in the library.

Table 11: Detailed description of tool definition fields.

| Field Name | Type | Description |
|---|---|---|
| name | String | The unique name of the tool. |

Table 11 continued from previous page

| Field Name | Type | Description |
|---|---|---|
| description | String | A human-readable summary of the tool's purpose and functionality. |
| parameters | Array | An array of objects, where each object defines an input parameter for the tool. This includes the parameter's name, type, description, and whether it is required. |
| returns | Array | An array of objects describing the possible return values from the tool. Each object specifies the name, type, and a description of the return value. |
| errors | Array | A list of possible errors that the tool might throw, including an error code and description for each. |
| dependencies | Array | A list of other tools that must be executed before this tool can run. |
| dependency_metadata | Object | Metadata related to the tool's dependencies, including its execution level and order in a workflow. |
| metadata | Object | General metadata about the tool, such as its category, operation type, and creation timestamp. |
| canonical_name | String | The official, unique name for the tool, used to resolve any aliases. |
| aliases | Array | A list of alternative names that refer to this tool. |
| differentiation | Object | An object containing details that distinguish this tool from others, including its unique purpose, key differentiators, and usage keywords. |
| original_description | String | The initial, unenhanced description of the tool. |
| differentiation_enhanced | Boolean | A flag indicating whether the tool's differentiation information has been enhanced by an LLM. |

### B.1.4 EXAMPLE OF A TOOL

We also provide an example to demonstrate what a tool might look like. Recall that a tool named `file_operations_reader` is required in executing the task example. Here we demonstrate its registry in the tool library.

```
"file_operations_reader": {
    "name": "file_operations_reader",
    "description": "This tool is specifically designed to read and
    retrieve data from files.",
    "parameters": [
      {
        "name": "source",
        "type": "string",
        "description": "Source location or identifier",
        "required": true,
        "default": null,
        "constraints": {}
      },
      {
        "name": "options",
        "type": "object",
        "description": "Additional options",
        "required": false,
        "default": {},
        "constraints": {}
      }
    ],
    "returns": [
```

```json
    {
      "name": "success",
      "type": "boolean",
      "description": "Whether operation succeeded"
    },
    {
      "name": "data",
      "type": "object",
      "description": "Retrieved or parsed data"
    },
    {
      "name": "metadata",
      "type": "object",
      "description": "Operation metadata"
    }
  ],
  "errors": [
    {
      "code": "INVALID_INPUT",
      "description": "Input validation failed"
    },
    {
      "code": "OPERATION_FAILED",
      "description": "Operation could not be completed"
    },
    {
      "code": "TIMEOUT",
      "description": "Operation timed out"
    },
    {
      "code": "FILE_NOT_FOUND",
      "description": "Specified file not found"
    },
    {
      "code": "PERMISSION_DENIED",
      "description": "Insufficient permissions"
    }
  ],
  "dependencies": [],
  "dependency_metadata": {
    "level": 0,
    "execution_order": 0,
    "category": "file_operations"
  },
  "metadata": {
    "category": "file_operations",
    "operation": "reader",
    "version": "1.0.0",
    "created_at": "2025-06-27T17:38:31.057340",
    "dependency_level": 0,
    "execution_order": 0
  },
  "canonical_name": "file_operations_reader",
},
```

In summary, the registration of a tool in the tool library, also often referred to as MCP (Model Context Protocol), is presented as a standardized interface system that provides structured access to external tools and services. Rather than a custom protocol implementation, it uses generated JSON schemas and client libraries that define how AI agents can interact with various tools. Specifically,

the schemas specify (1) the *input parameters* (given as `parameters` in the following) that an agent needs to provide to the tool, and (2) the possible outputs of executing the tool, including *return values* (given as `returns`) and *error messages* (given as `errors`). The schema also specifies other information, including the dependencies of the tool. If a tool has other dependencies, all dependencies must be successfully executed before the tool can be called.

Provided with the task to complete, and the tool library for usage, the LLM is required to perform tool calls, and check their return values to ensure task completion.

## B.2 PROMPTS

We provide the LLMs with multiple types of task-related *prompts*, carrying necessary information regarding the task. The LLMs are prompted to identify and execute tools to complete the task. The prompts of ResiliBench are classified into 4 categories:

1. Baseline prompts: Only contains basic information, such as task description, input and output, tool call instructions, etc.

2. CoT (Chain of Thought) prompts: Baseline prompts enhanced with CoT reasoning instructions.

3. MDP-optimal workflow prompts: Baseline prompts enhanced with a detailed workflow execution plan.

4. Flawed workflow prompts: Optimal prompts with various flaws injected to the workflow execution plan description.

### B.2.1 AN EXAMPLE OF BASELINE PROMPT

```
Execute a simple task.

Task: Leverage innovative multi-tool workflows to metamorphose
ambiguous input into strategic insights, enhancing operational
efficacy and fostering data-driven decisions, thus unlocking
untapped business potential through streamlined transformation.

Input Data:
- input_data: numeric array with 5 values

Expected Output:
- processed_data: processed data with success status

Tool Search Available:
You have access to a comprehensive tool library with specialized
tools for various operations.
To find relevant tools, use the search command: <tool_search>your
search query</tool_search>

Examples of tool searches:
- <tool_search>file reader writer</tool_search>
- <tool_search>data validation parser</tool_search>
- <tool_search>network api fetch</tool_search>

After finding the tools you need, execute them using:
<tool_call>tool_name</tool_call>

Instructions:
1. Analyze the task requirements
2. Search for appropriate tools based on what you need to do
3. Execute the tools in the correct order
4. Complete the task and indicate when finished
```

```
Use appropriate tools to complete the task.
```

### B.2.2 COT REASONING INSTRUCTIONS

```
**Think step by step about which tools to use and why.**

Please:
1. First explain your reasoning about which tools to use
2. Then execute the tools in the order you determined
3. Format tool calls as: <tool_call>tool_name</tool_call>

Begin with "Reasoning:" followed by your thought process.

Use appropriate tools to complete the task.
```

The instructions are directly appended to the baseline prompt to form a CoT prompt.

### B.2.3 AN EXAMPLE OF MDP-OPTIMAL WORKFLOW PROMPT

We provide and example of a workflow execution plan with summary:

```
## Workflow Execution Plan

1. Execute data_processing_transformer
   - Reason: Semantic match for 'full_description' operation
   - Requires: data_processing_parser
2. Execute data_processing_filter
   - Reason: Semantic match for 'full_description' operation
3. Execute data_processing_parser
   - Reason: Semantic match for 'full_description' operation
4. Execute computation_analyzer
   - Reason: Semantic match for 'full_description' operation
   - Requires: data_processing_parser, data_processing_aggregator
5. Execute file_operations_scanner
   - Reason: Semantic match for 'full_description' operation
6. Execute file_operations_reader
   - Reason: Semantic match for 'full_description' operation
7. Execute data_processing_validator
   - Reason: Semantic match for 'full_description' operation
   - Requires: data_processing_parser
8. Execute computation_calculator
   - Reason: Semantic match for 'full_description' operation
   - Requires: data_processing_parser, network_validator
9. Execute network_monitor
   - Reason: Semantic match for 'full_description' operation
10. Execute data_processing_aggregator
   - Reason: Semantic match for 'full_description' operation
   - Requires: data_processing_parser

### Analysis:
   - Critical tools identified: data_processing_filter,
   data_processing_transformer, data_processing_parser

### Execution Strategy:
   1. Follow the recommended sequence for optimal results
   2. Use alternatives if primary tools fail
```

```
    3. Pay special attention to critical tools
```

```
Use appropriate tools to complete the task.
```

The workflow execution plan is attached directly to the baseline prompt to form an MDP-optimal workflow prompt. In section D, we will explain how the MDP-optimal workflow prompt is constructed.

### B.2.4   FLAWED WORKFLOW PROMPT

There are several ways to perturb an MDP-optimal workflow execution plan to make it flawed. For example, we can introduce flaws to the order of the tool execution sequence. We can also introduce missing steps or redundancies. An ideally robust LLM is expected to discover and correct the flaws.

To test the capability of the LLM, we include Sequential Ordering Errors, where tools are executed in an incorrect sequence, and Tool Misuse Errors, which involve selecting an inappropriate tool that appears suitable. Parameter Configuration Errors occur from incorrect or omitted parameters, while Missing Critical Steps involves the strategic removal of essential operations. We also observe Redundant Operations, where unnecessary tools are added; Logic Discontinuity, where a tool's output is incompatible with the next tool's input; and Semantic Drift, a gradual deviation from the intended workflow caused by replacing tools with functionally different but semantically similar alternatives. We refer to C.3 for details of the flaws.

### B.3   TOOL RESULT SIMULATOR

Once the LLM performs a tool call, a simulator provides a return message that is either a success or a type of failure. For simplicity, we create a unified simulator for every tool, which calculates the success rate of a specific tool execution, then randomly samples a success or a failure.

The specific simulator implementation is decomposed as follows:

**Success rate calculator.**   The success rate calculator assigns a success rate $p \in [0, 1]$ for each specific tool execution. Specifically, the calculator first assigns a base success rate of $p_0 = 0.8$. Then, dependencies of the current tool is checked. For each dependency that has not been previously called, a penalty of $0.5$ is multiplied to the success rate. For each dependency that has been previously called but not successfully executed, a penalty of $0.7$ is multiplied to the success rate. Further, failure histories of the current task decreases the success rate of the current tool execution. Each history failure record induces a penalty of $0.9$ multiplied to the success rate. In summary, we have

$$\rho_{\text{success}} = \rho_{\text{base}} \cdot \prod 0.5^{N_u} \cdot 0.7^{N_f} \cdot 0.9^{N_h}$$

where $N_u$ is the number of dependencies not called, $N_f$ is the number of dependencies (called but) not successfully executed, and $N_h$ is the number of historical execution failures.

**Result simulator.**   Given the success rate $p$, the result simulator first randomly simulates a "success" or "failure". If the result is "success", the simulator creates the success-related return message provided in the tool's documentation. If the result is "failure", the simulator randomly selects a type of failure (e.g., TIMEOUT, EXECUTION_FAILURE) from the tool's documentation.

### B.4   TASK RESULT EVALUATION

The evaluation of task outcomes is performed by the InteractiveExecutor, which determines the success of a task based on a detailed analysis of the execution history. This section outlines the different levels of task success and the criteria used for their assessment.

### B.4.1 Levels of Task Success

**`full_success`:** This level indicates that the task was completed perfectly. It requires all of the specified tools to be executed successfully and in the correct sequence.

**`partial_success`:** This level represents a task that was completed to a significant extent but did not meet all the criteria for full success. This could mean, for example, that a majority of the required tools were executed, or that a valid output was generated despite some tools failing.

**`failure`:** This level is assigned when the task could not be meaningfully completed. This typically occurs if a critical number of tools fail, no output is generated for essential tasks, or the execution gets stuck in a loop.

### B.4.2 Evaluation Criteria

The executor evaluates the success of a task by assessing several key factors from the execution state. The final success level is determined by a combination of these criteria:

- **Required Tools Coverage:** This is the most critical metric. The evaluator checks what percentage of the tools listed in the task's `required_tools` list were executed successfully. Full success requires 100% coverage.

- **Sequence Correctness:** For a task to be considered a `full_success`, the required tools must not only be executed but also be called in the precise order specified in the task definition.

- **Output Generation:** The evaluator checks whether any tool that is expected to produce an output (e.g., tools with names like 'writer', 'exporter', 'saver') was successfully executed. The generation of an output is an indicator of full success and partial success, especially for pipeline-oriented tasks.

- **Explicit Completion Signal:** The conversation history is scanned for signals from the language model indicating that it considers the task complete (e.g., "task completed", "finished executing").

- **Minimum Tool Execution:** For each task type, there is a minimum number of successfully executed tools required to be considered for `partial_success`. For instance, an `advanced_computation_pipeline` requires more successful tool calls than a `simple_data_transformation`.

- **Termination Conditions:** The evaluation also considers reasons for premature termination, such as an excessive number of consecutive tool failures or getting stuck in a repetitive loop, which would typically lead to a `failure` rating.

A task is rated as a `partial_success` if it meets at least two of the conditions (e.g., has over `Minimum Tool Execution` tool coverage and generates an output). If the conditions for either full or partial success are not met, the task is marked as a `failure`.

## C  Benchmark Construction

In this section, we introduce how the tasks are automatically constructed. Specifically, this includes the generation of the *task documentation*, *tool library* and *prompts*.

### C.1  Task and Tool construction

The generation framework employs a *category-mediated* architecture where tool categories serve as the primary interface between tool capabilities and task requirements, ensuring that generated tasks are both reasonable and logically correspond to the underlying tool ecosystem.

### C.1.1  Tool Generation Methodology

We first generate the tool library, which is independent of tasks, and generated via a LLM-free rule-based process. This process highly relies on categories, where the basic layer of category consists

of a category-operation matrix consisting of 30 distinct tool types. Then, we introduce semantic operations and semantics matching that re-categorizes these tools, in order to (1) introduce aliases to flexibly create more tools and (2) facilitate and interface with task creation.

**Layer 1: Category-Operation Matrix (Tool Definition Layer).** The system defines 6 categories, each with 5 specific operations, creating a $6 \times 5$ matrix of 30 distinct tool types. The categories and their operations are given as

- data_processing: parser, transformer, validator, aggregator, filter
- file_operations: reader, writer, scanner, compressor, converter
- network: fetcher, poster, monitor, validator, router
- computation: calculator, analyzer, optimizer, simulator, predictor
- integration: connector, authenticator, mapper, queue, scheduler
- utility: logger, cache, notifier, tracker, helper

Within the matrix, each category-operation pair (e.g., data_processing_parser, network_fetcher) generates a unique tool with:

- Specific parameter templates
- Specific return value templates
- Specific error handling templates
- Category-specific behavior patterns

The category-operation matrix-based templates generate the following information of tools:

- **Naming Convention.** Tools follow a systematic `{category}_{operation}` pattern, ensuring consistent identification and categorization.
- **Parameter Assignment.** The system uses rule-based logic to assign appropriate parameters based on operation semantics. For example, reading operations (e.g., reader, parser, scanner) receive `source` parameters, and transformation (e.g., transformer, converter, mapper) operations receive `input_format` and `output_format` specification parameters.
- **Return Value Generation.** Return values are systematically assigned based on operation type. For example, operations for receiving data, e.g., reader, fetcher, scanner, returns `data` values. Other than the operation-specific return values, all tools provide success indicators and metadata.

  Another part of return values is the error information, where we also create general errors such as `INVALID_INPUT`, and operation-specific errors such as `OVERFLOW` for computation operations.

**Layer 2: Semantic Operation Types (Workflow/Dependency Layer).** The system then regroups the 30 operations into 5 semantic operation types based on their role in data processing workflows:

```python
operation_types = {
    'sources': ['reader', 'fetcher', 'scanner', 'authenticator'],
    'processors': ['parser', 'transformer', 'validator',
    'analyzer', 'calculator'],
    'aggregators': ['aggregator', 'combiner', 'merger'], # Note:
    some are aliased
    'outputs': ['writer', 'poster', 'exporter', 'notifier'], #
    Note: some are aliased
    'utilities': ['logger', 'cache', 'tracker', 'monitor']
}
```

We have the following remarks:

1. **Cross-category semantic grouping**. Operations from different categories can belong to the same semantic type. For example:
   - sources includes: reader (file_operations), fetcher (network), scanner (file_operations), authenticator (integration)
   - processors includes: parser (data_processing), transformer (data_processing), validator (data_processing/network), analyzer (computation), calculator (computation)

2. **Some operations have aliases**. The semantic layer introduces some operations not explicitly in the original 30:
   - combiner and merger (aliases for aggregator)
   - exporter (alias for writer or converter)

3. **Workflow-based logic**. This semantic grouping enables dependency rules like:

```
dependency_rules = {
    'processors': {
        'transformer': ['parser', 'reader'],  # transformers
        depend on parsers and readers
        'validator': ['parser', 'transformer'],
        'analyzer': ['parser', 'aggregator'],
        'calculator': ['parser', 'validator']
    },
    # ... more rules
}
```

Based on the dependency rules, the prerequisites of each tool is configured, facilitating construction of complicated workflows with dependencies.

**How the Two Layers Work Together.**   We now summarize how the two layers of categorization work together to generate tool information, and facilitate task construction. First, in the *tool generation phase*, we use Layer 1 (category + operation) to create specific tools. Each tool gets a unique name and a category-specific parameter/return template. Next, we move to the *dependency resolution phase*, where we use Layer 2 (semantic operation types) to determine logical dependencies.

Lastly, the *task construction phase*, which will be introduced shortly, combines both layers to build workflow-aware task templates. This phase uses semantic types to understand data flow patterns, and uses specific category-operation tools as the actual implementation.

## C.2    TASK GENERATION METHODOLOGY

This section describes a systematic approach for generating computational tasks based on a library of available tools, ensuring that generated tasks are both reasonable and logically correspond to the capabilities of the underlying tool ecosystem. The framework employs a multi-layered architecture that combines predefined task templates, semantic tool matching, and large language model (LLM) augmentation to create diverse, executable task instances.

**Foundational Task Categories**   The system first defines five types of tasks, including basic file processing, simple data transformation, complex validation pipeline, complex network integration, and advanced computation pipeline.

Then, for each category of tasks, we define a standard operation sequence, given as

```
operation_sequences = {
'basic_file_processing': ['input', 'process'],
'simple_data_transformation': ['read', 'process', 'output'],
'complex_validation_pipeline': ['read', 'validate',
'transform', 'aggregate', 'write'],
'complex_network_integration': ['fetch', 'parse', 'validate',
'transform', 'post'],
'advanced_computation_pipeline': ['read', 'validate',
'transform', 'compute', 'aggregate', 'write'],
}
```

The task generation process begins with a structured template system that defines the fundamental characteristics of different task types. Each task template serves as a blueprint that specifies the following core components:

- **Task requirements**: A `TaskRequirement` object that defines the minimum number of tools needed, required tool operations (such as "reader", "transformer", "writer"), and complexity constraints. This ensures that generated tasks align with available tool capabilities and maintain logical coherence.
- **Template structure**: Each template includes a task type identifier, descriptive requirements, and objectives that guide the generation process. The template acts as a constraint mechanism, ensuring that only feasible task combinations are considered.
- **Tool category analysis**: The system performs automated analysis of the available tool library to identify tool categories and their associated operations. This analysis informs template creation, ensuring that templates are grounded in actual tool availability rather than abstract specifications.

**Semantic Tool Matching.** To connect tasks to the pre-generated tool system, the framework incorporates semantic matching to select appropriate tools for task generation. For example, when `read` exists in the operation sequence of a task, RAG semantic matching would lead to operations such as `reader`, `fetcher` and `scanner` in the tool *semantic operation types*. This further leads to retrieving specific tools, such as `file_operations_reader`. To create variations in one task category, the specific tool matching with a task operation has multiple choices (e.g., read can correspond to `file_operations_reader` or `network_fetcher`). For each task, only one specific tool is selected for each operation step. Other sources of variation come from the choice of different templates.

**Task Construction Example.** We now provide a simple example of how a task might be constructed.

Step 1: Choice of tools

```
complex_validation_pipeline = ['read', 'validate', 'transform',
'aggregate', 'write']

# semantics search result:
'read' -> choose 'file_operations_reader'
'validate' -> choose 'data_processing_validator'
'transform' -> choose 'data_processing_transformer'
'aggregate' -> choose 'data_processing_aggregator'
'write' -> choose 'file_operations_writer'

# final required_tools = [a list of tools after dependency-based
sorting]
```

Step 2: Task construction

```
# 1. Task ID
instance_id = f"task_{uuid.uuid4().hex[:8]}"

# 2. Task description (template-based)
description = f"... #{i+1}"

# 3. Input and output (based on template and tools)
requirements = [
    TaskRequirement("processing_config", "Configuration for
    processing"),
    ...
```

```
]

objectives = [
    TaskObjective(
        "...",
        [f"Execute {tool}" for tool in selected_tools],
        "output"
    )
]

# 4. Difficulty (based on task type and tools)
complexity = "easy" if len(tools) <= 2 else "medium"
```

**LLM Enhancement.**  After the basic RAG tool-selection results of a task, we can also use LLM APIs to create an instance of a task.

Specifically, we provide to the LLM related RAG semantic search result together with the following prompt:

```
    prompt = f"""
You are an expert workflow designer. Based on RAG search results,
design {task_desc}.

Available tools (sorted by relevance):
{json.dumps(tools_info, indent=2)}

Requirements:
- Task type: {task_type}
- Complexity: {complexity}
- Select 3-6 most appropriate tools based on:
  1. RAG relevance scores
  2. Logical workflow sequence
  3. Tool categories and operations
  4. Input/output compatibility between tools

Design a complete task instance with:
1. A clear description of what the task accomplishes
2. A logical sequence of tools that work together
3. Realistic input data that matches the first tool's parameters
4. Expected output that matches the final tool's returns
"""
```

The LLM enhancement provides a more natural and detailed task description, and ensures consistency (e.g., among choices of tools and description) within the task setting.

## C.3 WORKFLOW PROMPT GENERATION

Each task has several corresponding prompts that guide the tested LLM to complete the task. The baseline prompts and CoT prompts can be directly transformed from the task profile in the generated task library. The MDP-optimal workflow prompt is generated through MDP training (details provided in Section D), while the flawed workflow prompt is created by inserting deliberate flaws.

We present a comprehensive framework for systematically generating flawed workflows from optimal sequences. Our approach introduces seven distinct categories of workflow defects with varying severity levels, enhanced by semantic similarity detection through RAG techniques.

- **Sequential Ordering Errors (Order Flaws)**. Sequential ordering errors occur when tools are executed in incorrect or flawed sequences, violating logical dependencies or temporal constraints.

To inject this error into the workflow, we use the *swap method*, i.e., creating random permutations of adjacent tool pairs. We also use the *dependency violation method*, which repositions dependent tools before their prerequisites, to inject more logical flaws.

- **Tool Misuse Errors** Tool misuse represents the selection of inappropriate tools that appear suitable but lack the required functionality for the specific context.

  To inject this error, we use the *semantic similarity method*. leveraging RAG-based semantic search to identify tools with similar descriptions but different functionalities. We also use *category mismatch method*, which replaces tools with alternatives from entirely different functional categories, simulating gross misunderstanding of tool capabilities.

- **Parameter Configuration Errors**. Parameter errors involve incorrect specification or omission of required tool parameters.

  To inject this error, we use the *missing parameters method*, which systematically removes required parameters based on tool specifications. We also use the *type mismatch method*, which introduces parameters with incorrect data types or value ranges. Parameter error injection follows the formula $P_{error} = \alpha \times P_{required} + \beta \times P_{optional}$, where $\alpha$ and $\beta$ are severity-dependent coefficients.

- **Missing Critical Steps**. Workflow incompleteness through strategic removal of essential tools or validation steps.

  To inject this error, we use the *middle step removal method*, which eliminates intermediate processing steps while preserving workflow endpoints. We also use the *validation removal method*, which systematically removes quality assurance and verification tools.

- **Redundant Operations**. Introduction of unnecessary or duplicated operations that increase computational overhead without adding value.

  To inject this error, we use the *duplication method*, which repeats existing tools within the sequence. We also use the *unnecessary addition method*, which inserts tools that provide no functional benefit to the task.

- **Logic Discontinuity**. Breaks in logical flow where tool outputs become incompatible with subsequent tool inputs.

  To inject this error, we use the *format mismatch method*, which introduces incompatible data format transitions between tools. We also use the *unrelated insertion method*, which adds tools unrelated to the primary task objective.

- **Semantic Drift (RAG-Enhanced)**. Advanced error patterns enabled by semantic understanding, representing gradual deviation from intended workflow semantics.

  To inject this error, we use the *semantic mismatch method*, which replaces tools with semantically similar but functionally inappropriate alternatives. We also use the *semantic drift method*, which performs progressive replacement where each subsequent tool is selected based on the previous replacement, creating cumulative semantic deviation. The drift function is defined as $T_i = \arg\max_{t \in T}(S(t, T_{i-1}) \times (1 - F(t, T_{original})))$, where $S$ represents semantic similarity and $F$ represents functional equivalence.

# D MDP-BASED WORKFLOW GENERATION TRAINING MECHANISM

This section provides a comprehensive analysis of the Markov Decision Process (MDP) training framework used for MDP-optimal workflow generation in our system.

## D.1 MDP STATE SPACE FORMULATION

The MDP state space $\mathcal{S}$ is defined by a composite representation that captures both task semantics and execution dynamics. Each state $s_t \in \mathcal{S}$ is represented as:

$$s_t = \langle \tau, \omega; \psi, \phi, \xi \rangle$$

where $\tau$ and $\omega$ are task-invariant components that remain constant throughout execution, while $\psi$, $\phi$, and $\xi$ capture the dynamic execution state.

**Static Task Components** $(\tau, \omega)$**:** The task identification $\tau = \{\text{task\_id}, \text{task\_type}, \text{task\_objective}\}$ and semantic feature vector $\omega \in \mathbb{R}^{20}$ encode task requirements (input/output needs, domain type, complexity level) as normalized features. These components are extracted once at initialization and provide consistent task context for all subsequent decisions.

**Tool Execution States Component** $(\psi)$

$$\psi = \{\sigma_i : i \in \mathcal{T}\} \text{ where } \sigma_i \in \Sigma$$

The tool execution status set $\Sigma$ contains:

$$\Sigma = \{\text{NOT\_ATTEMPTED}, \text{QUEUED}, \text{RUNNING}, \text{SUCCESS},$$
$$\text{FAILED}, \text{TIMEOUT}, \text{DEPENDENCY\_FAILED}\}$$

where $\mathcal{T}$ represents the available tool set and each $\sigma_i$ tracks the execution status of tool $i$.

**Progress and Workflow Component** $(\phi)$

$$\phi = \{p, k, \mathcal{M}_{\text{achieved}}, \mathcal{M}_{\text{expected}}, \mathbf{e}_{\text{seq}}\}$$

where:

- $p \in [0, 1]$ denotes overall task progress
- $k \in \mathbb{N}$ represents the current workflow step
- $\mathcal{M}_{\text{achieved}} \subseteq \mathcal{M}_{\text{expected}}$ are milestone sets
- $\mathbf{e}_{\text{seq}} = [\iota_1, \iota_2, \ldots, \iota_k]$ is the tool execution sequence

**RAG-Enhanced Context Component** $(\xi)$ This component integrates retrieval-augmented generation capabilities:

$$\xi = \{\mathcal{R}_{\text{rag}}, \mathbf{E}_{\text{cache}}, \mathbf{C}_{\text{semantic}}, \mathbf{H}_{\text{selection}}, \mathcal{T}_{\text{candidates}}\}$$

where:

- $\mathcal{R}_{\text{rag}}$: RAG search results mapping semantic operations to tool-score pairs
- $\mathbf{E}_{\text{cache}}$: Pre-computed embedding similarity scores for all tools
- $\mathbf{C}_{\text{semantic}}$: Semantic confidence scores per tool
- $\mathbf{H}_{\text{selection}}$: Tool selection history with metadata
- $\mathcal{T}_{\text{candidates}}$: Capability-type to candidate-tools mapping for fallback selection

**State Encoding for Neural Networks** The composite state is encoded into a fixed-size vector $\mathbf{s}_t \in \mathbb{R}^d$ through concatenation and normalization:

$$\mathbf{s}_t = \text{Concat}(\mathbf{encode}(\psi), \mathbf{encode}(\phi), \mathbf{encode}(\xi), \omega)$$

where $\mathbf{encode}(\cdot)$ transforms discrete components into continuous representations using one-hot encoding, progress normalization, and embedding lookups.

This multi-faceted state representation enables the MDP to capture both structural workflow dependencies and semantic task requirements, facilitating more informed action selection in complex tool orchestration scenarios.

### D.2 MDP ACTION SPACE DEFINITION

The action space $\mathcal{A}$ consists of structured actions that operate on tool orchestration and workflow management. Each action $a \in \mathcal{A}$ is represented as a composite tuple:

$$a = \langle \alpha, \iota, \beta, \theta, c \rangle$$

where the component $\alpha$ refers to an action (with respect to the tool), $t$ is the specific choice of target tool, and $\beta$ is the consequent relevant information. Specifically, we introduce the components as follows:

**Action Type Component ($\alpha$)**  The action type $\alpha$ belongs to a finite set of predefined operations:

$$\alpha \in \mathcal{A}_{\text{type}} = \{\texttt{INVOKE\_TOOL}, \texttt{VALIDATE\_OUTPUT}, \texttt{RETRY\_TOOL},$$
$$\texttt{RECOVER\_ERROR}, \texttt{CHECK\_DEPENDENCIES},$$
$$\texttt{CREATE\_CHECKPOINT}, \texttt{RESTORE\_CHECKPOINT},$$
$$\texttt{NO\_OP}, \texttt{PARALLEL\_EXECUTE}\}$$

**Tool Target Component ($\iota$)**  The tool identifier $\iota \in \mathcal{T} \cup \{\emptyset\}$ specifies the target tool for execution, where $\mathcal{T}$ represents the available tool set and $\emptyset$ indicates no specific tool target.

**Semantic Enhancement Component ($\beta$)**  The RAG-enhanced component $\beta = \langle s_{\text{sem}}, \mathcal{S}_{\text{src}}, \mathcal{T}_{\text{alt}} \rangle$ contains:

- $s_{\text{sem}} \in [0, 1]$: Semantic relevance score from embedding-based search
- $\mathcal{S}_{\text{src}} \in \{\text{rule}, \text{embedding}, \text{hybrid}, \text{pattern}\}$: Information source type
- $\mathcal{T}_{\text{alt}} \subseteq \mathcal{T}$: Alternative tool candidates

**Parameters Component ($\theta$)**  Action parameters $\theta$ contain execution-specific configuration as a key-value mapping, enabling flexible parameterization for different action types.

**Confidence Component ($c$)**  The confidence score $c \in [0, 1]$ represents the system's belief in the action's appropriateness for the current state, computed through multi-factor assessment.

### D.2.1 ACTION FILTERING MECHANISM

Valid actions at state $s_t$ are determined through a multi-stage filtering process $\mathcal{F} : \mathcal{S} \to 2^{\mathcal{A}}$:

$$\mathcal{A}_{\text{valid}}(s_t) = \mathcal{F}_{\text{semantic}} \circ \mathcal{F}_{\text{dependency}} \circ \mathcal{F}_{\text{constraint}}(\mathcal{A})$$

where:

**Constraint Filter ($\mathcal{F}_{\text{constraint}}$):** Removes actions violating basic execution constraints:
$$\mathcal{F}_{\text{constraint}}(\mathcal{A}) = \{a \in \mathcal{A} : \text{status}(a.\iota) \neq \texttt{SUCCESS} \wedge$$
$$\text{retries}(a.\iota) < 3 \wedge$$
$$\text{parallel\_safe}(a.\iota) \vee |\text{running\_tools}(s_t)| = 0\}$$

Here $a.\iota$ represents the choice of tool $\iota$ of action $a$.

**Dependency Filter ($\mathcal{F}_{\text{dependency}}$):** Ensures prerequisite tools are successfully executed:
$$\mathcal{F}_{\text{dependency}}(\mathcal{A}') = \{a \in \mathcal{A}' : \forall d \in \text{deps}(a.\iota), \psi_d = \texttt{SUCCESS}\}$$

**Semantic Filter ($\mathcal{F}_{\text{semantic}}$):** Applies multi-source confidence scoring:
$$\mathcal{F}_{\text{semantic}}(\mathcal{A}'') = \{a \in \mathcal{A}'' : c_{\text{composite}}(s_t, a) > \tau_{\text{threshold}}\}$$

The composite confidence combines multiple information sources:

$$c_{\text{composite}}(s_t, a) = \sum_i w_i \cdot c_i(s_t, a) \tag{1}$$

with weights $\mathbf{w} = [w_{\text{rule}}, w_{\text{embed}}, w_{\text{pattern}}, w_{\text{task}}] = [0.25, 0.30, 0.25, 0.20]$ and individual confidence components:

$$c_{\text{rule}}(s_t, a) = f_{\text{alignment}}(\omega.f_{\text{task}}, \text{ops}(a.\iota))$$
$$c_{\text{embed}}(s_t, a) = \xi.\mathbf{E}_{\text{cache}}[a.\iota]$$
$$c_{\text{pattern}}(s_t, a) = \max_{p \in \mathcal{P}} p_{\text{score}} \cdot \mathbf{1}[a.\iota \text{ extends } p]$$
$$c_{\text{task}}(s_t, a) = \text{preference}(\tau.\text{task\_type}, a.\iota)$$

### D.2.2 FALLBACK ACTION GENERATION

When $|\mathcal{A}_{\text{valid}}(s_t)| \leq 1$ (only NO_OP available), the system employs progressive fallback strategies:

---

**Algorithm 1** Fallback Action Generation

---

1: **if** $\xi.\mathcal{T}_{\text{candidates}} \neq \emptyset$ **then**
2:     Add top-2 RAG candidates with $c = 0.35$
3: **else if** workflow_step $< 10$ **then**
4:     Add dependency-free tools with $c = 0.30$
5: **else**
6:     Force recovery actions: $\alpha \in \{\text{RECOVER\_ERROR}, \text{RESTORE\_CHECKPOINT}\}$
7: **end if**

---

### D.2.3 ACTION SPACE CARDINALITY

The total action space size scales as:

$$|\mathcal{A}| = |\mathcal{A}_{\text{type}}| \times (|\mathcal{T}| + 1) + |\mathcal{A}_{\text{meta}}|$$

where $|\mathcal{A}_{\text{meta}}| = 4$ represents tool-independent actions (NO_OP, CREATE_CHECKPOINT, etc.). However, the effective action space at any state $s_t$ is typically $|\mathcal{A}_{\text{valid}}(s_t)| \ll |\mathcal{A}|$ due to filtering constraints.

This structured action representation enables the MDP to maintain semantic coherence while providing sufficient flexibility for complex workflow orchestration scenarios.

## D.3 MULTI-PHASE ADAPTIVE REWARD FUNCTION

The reward function implements a sophisticated two-phase training strategy that adapts based on the agent's learning progress. This design addresses the fundamental challenge of learning both *what* tools to use and *when* to use them.

### D.3.1 PHASE-ADAPTIVE STRATEGY

The core insight is that tool selection learning requires different optimization objectives at different stages:

$$R(s_t, a_t, s_{t+1}) = \begin{cases} R_{\text{coverage}}(s_t, a_t, s_{t+1}) & \text{if } \rho_{\text{success}} < \theta_{\text{adapt}} \\ R_{\text{sequence}}(s_t, a_t, s_{t+1}) & \text{otherwise} \end{cases}$$

where $\rho_{\text{success}}$ represents the current success rate and $\theta_{\text{adapt}} = 0.3$ is the adaptation threshold.

**Phase I: Coverage-Focused Learning ($\rho_{\textbf{success}} < 0.3$)**

During initial learning, the agent must discover which tools are relevant for different task types. The reward function prioritizes exploration and tool discovery:

$$R_{\text{coverage}} = R_{\text{exploration}} + R_{\text{discovery}} + R_{\text{completion}}$$
$$R_{\text{exploration}} = \alpha_{\text{attempt}} \cdot \mathbf{1}[\text{tool attempted}] + \alpha_{\text{novel}} \cdot \mathbf{1}[\text{first attempt}]$$
$$R_{\text{discovery}} = \beta_{\text{required}} \cdot \mathbf{1}[\iota \in \mathcal{T}_{\text{required}}] \cdot \mathbf{1}[\text{success}]$$
$$R_{\text{completion}} = \gamma_{\text{progress}} \cdot \Delta p + \delta_{\text{milestone}} \cdot |\mathcal{M}_{t+1} - \mathcal{M}_t|$$

with reward weights: $\alpha_{\text{attempt}} = 3$, $\alpha_{\text{novel}} = 30$, $\beta_{\text{required}} = 50$, $\gamma_{\text{progress}} = 100$, $\delta_{\text{milestone}} = 40$.

The key insight is that *any* successful execution of required tools receives substantial rewards, regardless of execution order.

**Phase II: Sequence-Optimized Learning ($\rho_{\text{success}} \geq 0.3$)**

Once basic tool usage is learned, the focus shifts to optimizing execution sequences and workflow efficiency:

$$R_{\text{sequence}} = R_{\text{order}} + R_{\text{efficiency}} + R_{\text{completion}}$$

$$R_{\text{order}} = \begin{cases} 15 \cdot |\mathcal{T}_{\text{correct\_seq}}| & \text{if perfect sequence order} \\ 5 \cdot |\mathcal{T}_{\text{near\_seq}}| & \text{if near-correct order} \\ -5 \cdot \sum_\iota |i_{\text{actual}}(\iota) - i_{\text{expected}}(\iota)| & \text{otherwise} \end{cases}$$

$$R_{\text{efficiency}} = \eta_{\text{step}} \cdot \frac{\max(0, k_{\text{target}} - k)}{k_{\text{target}}} + \eta_{\text{error}} \cdot \mathbf{1}[e_{\text{total}} = 0]$$

where $i_{\text{actual}}(\iota)$ and $i_{\text{expected}}(\iota)$ represent actual and expected positions of tool $\iota$ in the execution sequence.

### D.3.2 UNIVERSAL REWARD COMPONENTS

Several reward components operate consistently across both phases:

**Progress Incentives:**

$$R_{\text{progress}} = \gamma_{\text{base}} \cdot \Delta p + \gamma_{\text{early}} \cdot \Delta p \cdot \mathbf{1}[k < 10] + \gamma_{\text{late}} \cdot \Delta p \cdot \mathbf{1}[p > 0.8]$$

**RAG-Enhanced Semantic Alignment:**

$$R_{\text{semantic}} = \lambda_{\text{rag}} \cdot s_{\text{rag}}(\iota) + \lambda_{\text{pattern}} \cdot s_{\text{pattern}}(\mathbf{e}_{\text{seq}}, \iota) + \lambda_{\text{task}} \cdot s_{\text{task}}(\tau, \iota)$$

where $s_{\text{rag}}(\iota)$ is the RAG similarity score, $s_{\text{pattern}}$ captures learned sequential patterns, and $s_{\text{task}}$ represents task-tool alignment.

**Terminal Rewards:**

Upon episode completion, substantial rewards are distributed based on both success and training phase:

$$R_{\text{terminal}} = \begin{cases} 150 + 50 \cdot r_{\text{coverage}} & \text{if coverage phase and success} \\ 100 + 50 \cdot r_{\text{sequence}} & \text{if sequence phase and success} \\ \max(0, 50 \cdot p_{\text{final}}) & \text{if failure} \end{cases}$$

where $r_{\text{coverage}} = \frac{|\mathcal{T}_{\text{executed}} \cap \mathcal{T}_{\text{required}}|}{|\mathcal{T}_{\text{required}}|}$ and $r_{\text{sequence}}$ measures sequence correctness.

### D.3.3 PENALTY STRUCTURE

The penalty system is designed to be *adaptive* and *minimal* during early learning:

$$R_{\text{penalty}} = R_{\text{error}} + R_{\text{repetition}} + R_{\text{stagnation}}$$

$$R_{\text{error}} = \begin{cases} 0 & \text{if } \rho_{\text{success}} < 0.1 \\ -2 \cdot \Delta e_{\text{total}} & \text{if } 0.1 \leq \rho_{\text{success}} < 0.3 \\ -5 \cdot \Delta e_{\text{total}} & \text{otherwise} \end{cases}$$

$$R_{\text{repetition}} = -\kappa \cdot (\text{count}(\iota, \mathbf{e}_{\text{seq}}) - 1) \cdot \mathbf{1}[\text{count}(\iota) \geq 2]$$

$$R_{\text{stagnation}} = -10 \cdot \mathbf{1}[a_t = a_{t-1} = \text{NO\_OP}]$$

where $\kappa = 1$ during coverage phase and $\kappa = 5$ during sequence phase.

### D.3.4 KEY DESIGN PRINCIPLES

This reward architecture embodies several key principles:

- **Progressive Complexity**: Early learning focuses on tool discovery; later learning optimizes execution patterns
- **Semantic Guidance**: RAG-enhanced rewards align tool selection with task semantics
- **Adaptive Penalties**: Error tolerance decreases as competence increases
- **Terminal Differentiation**: Final rewards depend on both success and current learning objectives

The phase transition at $\rho_{\text{success}} = 0.3$ ensures that agents first master basic tool usage before attempting to optimize execution sequences, leading to more stable and efficient learning convergence.

## D.4 MDP ENVIRONMENT DYNAMICS

This section details the core environment mechanics that govern state transitions, completion criteria, and adaptive behavior in our MDP framework. Unlike conventional MDP environments with static rules, our system implements dynamic thresholds and curriculum-adaptive mechanisms that evolve during training.

### D.4.1 ENVIRONMENT STEP EXECUTION

The environment step function orchestrates the complete MDP transition process through three sequential phases:

$$\text{MDPStep}(s_t, a_t) = \langle s_{t+1}, r_t, \text{done} \rangle$$

where each component is computed through dedicated sub-procedures:

$$s_{t+1} = \text{StateTransition}(s_t, a_t)$$
$$r_t = \text{AdaptiveReward}(s_t, a_t, s_{t+1})$$
$$\text{done} = \text{CompletionCheck}(s_{t+1})$$

**Phase I: State Transition ($s_{t+1} = \text{StateTransition}(s_t, a_t)$)**

The state transition simulates tool execution through an adaptive reliability model that adjusts success probability based on training progress:

$$\rho_{\text{success}}(\iota, s_t) = \rho_{\text{base}} \cdot \prod_{d \in \text{deps}(\iota)} \gamma_{\text{dep}} \cdot \gamma_{\text{retry}}^{r_\iota} \cdot \gamma_{\text{semantic}}$$

$$\rho_{\text{base}} = \begin{cases} 0.95 & \text{if } \rho_{\text{episode}} < 0.1 \\ 0.90 & \text{if } 0.1 \le \rho_{\text{episode}} < 0.3 \\ 0.85 & \text{otherwise} \end{cases}$$

where $\rho_{\text{episode}}$ represents the current episode success rate, $\gamma_{\text{dep}} = 0.8$ penalizes unmet dependencies, $\gamma_{\text{retry}} = 0.95$ reduces reliability with retry attempts $r_\iota$, and $\gamma_{\text{semantic}} \in [1.0, 1.3]$ rewards semantic task-tool alignment. We note that this differs from the tool result simulator probability settings B.3, which enables a testing environment different from the training environment.

The transition updates multiple state components:

- Tool execution states: $\psi_\iota^{t+1} \sim \text{Bernoulli}(\rho_{\text{success}}(\iota, s_t))$
- Execution sequence: $\mathbf{e}_{\text{seq}}^{t+1} = \mathbf{e}_{\text{seq}}^t \cup \{\iota\}$ if successful

- Error tracking: $e_{\text{consecutive}}^{t+1}$, $e_{\text{total}}^{t+1}$ updated based on outcome
- Data flow state progression: $\xi_{\text{flow}}^{t+1}$ advanced according to semantic operations

**Phase II: Progress Computation ($p_{t+1}$ within $s_{t+1}$)**

Progress update adapts to task structure through dual calculation modes:

$$p_{t+1} = \begin{cases} \frac{|\mathcal{T}_{\text{executed}} \cap \mathcal{T}_{\text{required}}|}{|\mathcal{T}_{\text{required}}|} + \beta_{\text{sequence}} & \text{if } \mathcal{T}_{\text{required}} \neq \emptyset \\ 0.3 \cdot p_{\text{milestone}} + 0.4 \cdot p_{\text{tool}} + 0.3 \cdot p_{\text{step}} & \text{otherwise} \end{cases}$$

where $\beta_{\text{sequence}} = 0.1$ provides sequence order rewards, $p_{\text{milestone}} = \frac{|\mathcal{M}_{\text{achieved}}|}{|\mathcal{M}_{\text{expected}}|}$, $p_{\text{tool}} = \min(1.0, \frac{|\text{successful\_tools}|}{5})$, and $p_{\text{step}} = \min(1.0, \frac{k}{20})$.

Progress is monotonically increasing: $p_{t+1} = \max(p_t, p_{\text{computed}})$ to prevent regression.

**Phase III: Reward Calculation ($r_t = \textbf{AdaptiveReward}(s_t, a_t, s_{t+1})$)**

The adaptive reward system operates through the two-phase strategy detailed in Section D.3, incorporating:

- Base exploration rewards for any non-NO_OP action
- Tool execution rewards scaled by training phase
- Progress increment rewards: $100 \cdot (p_{t+1} - p_t)$
- Required tool coverage bonuses (coverage phase) or sequence order bonuses (sequence phase)
- Semantic alignment rewards from RAG-enhanced action selection

**Phase IV: Completion Assessment (done $= \textbf{CompletionCheck}(s_{t+1})$)**

Episode termination employs hierarchical completion criteria:

$$\begin{aligned} \text{done} = \ &(p_{t+1} \geq 1.0) \vee \\ &(\mathcal{T}_{\text{required}} \subseteq \mathcal{T}_{\text{executed}}) \vee \\ &(k > k_{\max}) \vee \\ &(e_{\text{consecutive}} > e_{\max}) \vee \\ &(|\mathcal{A}_{\text{valid}}(s_{t+1})| \leq 1) \end{aligned}$$

where $k_{\max}$ and $e_{\max}$ are curriculum-adaptive thresholds. Success determination considers both completion method and achieved progress:

$$\text{success} = \begin{cases} \text{True} & \text{if } p_{t+1} \geq 0.95 \vee \mathcal{T}_{\text{required}} \subseteq \mathcal{T}_{\text{executed}} \\ p_{t+1} \geq 0.5 & \text{if timeout or deadlock} \\ \text{False} & \text{otherwise} \end{cases}$$

This four-phase execution framework ensures consistent state evolution while maintaining curriculum-appropriate difficulty and comprehensive performance assessment.

### D.4.2 CURRICULUM-ADAPTIVE TRAINING

Additionally, the training process implements dynamic curriculum adjustment based on performance metrics:

**Curriculum Stage Transitions:**

$$\text{stage}_{i+1} = \begin{cases} \text{stage}_i + 1 & \text{if } \rho_{\text{success}}^{(n)} > \theta_{\text{advance}} \wedge n \geq n_{\min} \\ \max(0, \text{stage}_i - 1) & \text{if } \rho_{\text{success}}^{(n)} < \theta_{\text{regress}} \\ \text{stage}_i & \text{otherwise} \end{cases}$$

where $i$ indexes curriculum update events, and $\rho_{\text{success}}^{(n)}$ represents the success rate over the last $n$ episodes. We have advancement threshold $\theta_{\text{advance}} = 0.7$, regression threshold $\theta_{\text{regress}} = 0.3$, and minimum episodes $n_{\min} = 50$.

Each curriculum stage modifies completion criteria given in table 12.

Table 12: Curriculum-dependent completion thresholds.

| Stage | Min Progress | Max Errors | Required Coverage |
|-------|--------------|------------|-------------------|
| 0 | 0.1 | 50 | 0.2 |
| 1 | 0.3 | 30 | 0.4 |
| 2 | 0.5 | 20 | 0.7 |
| 3 | 0.7 | 15 | 0.9 |
| 4+ | 0.7 | 10 | 1.0 |

### D.5 POLICY OPTIMIZATION FRAMEWORK AND MDP TRAINING

The MDP training implements a sophisticated optimization framework that combines policy gradient methods with curriculum learning and RAG-enhanced decision making. The optimization process operates through carefully orchestrated phases that build upon the adaptive reward structure. The policy update process operates directly on the multi-source confidence composition framework established earlier, optimizing the neural network that learns to weight and combine different information sources for action selection.

**Policy Architecture and Parameters:**

The trainable policy $\pi_\theta$ is implemented as a neural network that takes the encoded state representation $\mathbf{s}_t \in \mathbb{R}^d$ and outputs action probabilities over the filtered valid action space $\mathcal{A}_{\text{valid}}(s_t)$.

Critically, the policy network learns to implicitly weight the confidence components defined earlier:

$$c_{\text{learned}}(s_t, a_\iota) = \pi_\theta(a_\iota | s_t) \approx f_\theta \left( \sum_j w_j \cdot c_j(s_t, \iota) \right)$$

where $c_j \in \{c_{\text{rule}}, c_{\text{embed}}, c_{\text{pattern}}, c_{\text{task}}\}$ are the confidence components defined in Section D.2.1, and $f_\theta$ represents the learned non-linear transformation.

**Objective Function Correspondence:**

The optimization objective directly corresponds to maximizing the expected cumulative reward defined in Section D.3.1:

$$J(\theta) = \mathbb{E}_{\tau \sim \pi_\theta} \left[ \sum_{t=0}^{T} R_{\text{adaptive}}(s_t, a_t, s_{t+1}) \right]$$

$$= \mathbb{E}_{\tau \sim \pi_\theta} \left[ \sum_{t=0}^{T} \left( R_{\text{coverage}}(s_t, a_t, s_{t+1}) \cdot \mathbf{1}[\rho_{\text{success}} < 0.3] + R_{\text{sequence}}(s_t, a_t, s_{t+1}) \cdot \mathbf{1}[\rho_{\text{success}} \geq 0.3] \right) \right]$$

where $R_{\text{coverage}}$ and $R_{\text{sequence}}$ are the phase-specific reward functions detailed in equations (8)-(12) of Section 4.3.

#### D.5.1 EPISODE REWARD ADJUSTMENT

Post-episode reward adjustment propagates final performance back through the trajectory:

$$r'_t = r_t \cdot \mu_{\text{performance}} + \lambda_{\text{position}} \cdot (1 - \frac{t}{T}) + \delta_{\text{rag}} + \delta_{\text{pattern}}$$

$$\mu_{\text{performance}} = \begin{cases} 1.5 & \text{if score}_{\text{final}} \geq 0.9 \\ 1.2 & \text{if score}_{\text{final}} \geq 0.7 \\ 1.0 & \text{if score}_{\text{final}} \geq 0.5 \\ 0.7 & \text{otherwise} \end{cases}$$

where $\delta_{\text{rag}}$ rewards RAG-guided selections and $\delta_{\text{pattern}}$ rewards pattern completion.

**Gradient Computation with Experience Replay:**

The policy gradient incorporates the episode reward adjustment mechanism:

$$\nabla_\theta J(\theta) = \mathbb{E}_{\tau \sim \pi_\theta} \left[ \sum_{t=0}^{T} \nabla_\theta \log \pi_\theta(a_t|s_t) \cdot r'_t \right]$$

$$r'_t = r_t \cdot \mu_{\text{performance}} + \lambda_{\text{position}} \cdot (1 - \frac{t}{T}) + \delta_{\text{rag}} + \delta_{\text{pattern}}$$

where:

- $r_t = R_{\text{adaptive}}(s_t, a_t, s_{t+1})$ is the immediate reward from Section D.3.1
- $\mu_{\text{performance}} \in \{0.7, 1.0, 1.2, 1.5\}$ scales based on final episode score
- $\delta_{\text{rag}}$ provides additional reward when actions align with RAG-enhanced confidence $c_{\text{embed}}(s_t, \iota)$
- $\delta_{\text{pattern}}$ rewards actions that complete successful sequential patterns from $\mathcal{P}_{\text{successful}}$

**Curriculum-Adaptive Learning Rate:**

The learning rate adapts based on curriculum stage and phase transition status:

$$\alpha_{\text{policy}} = \alpha_{\text{base}} \cdot \gamma_{\text{curriculum}}^{\text{stage}} \cdot \beta_{\text{phase}}$$

$$\beta_{\text{phase}} = \begin{cases} 1.2 & \text{if recently transitioned to sequence phase} \\ 1.0 & \text{if stable in current phase} \\ 0.8 & \text{if performance declining} \end{cases}$$

where $\alpha_{\text{base}} = 3 \times 10^{-4}$ and $\gamma_{\text{curriculum}} = 0.95$.

**Integration with State Encoding:**

The policy update directly operates on the state encoding defined in Section 4.1:

$$\mathbf{s}_t = \text{Concat}(\mathbf{encode}(\psi_t), \mathbf{encode}(\phi_t), \omega.f_{\text{task}}, \mathbf{encode}(\xi_t))$$

$$\text{Loss}(\theta) = -\sum_{t=0}^{T} \log \pi_\theta(a_t|\mathbf{s}_t) \cdot r'_t + \beta_{\text{entropy}} \mathcal{H}(\pi_\theta(\cdot|\mathbf{s}_t))$$

where $\mathcal{H}(\cdot)$ is the entropy term encouraging exploration, with $\beta_{\text{entropy}} = 0.01$ during coverage phase and $\beta_{\text{entropy}} = 0.005$ during sequence phase.

**Convergence and Performance Monitoring:**

Policy convergence is monitored through multiple metrics aligned with the curriculum progression:

- Coverage Phase: Convergence measured by $\frac{|\mathcal{T}_{\text{executed}} \cap \mathcal{T}_{\text{required}}|}{|\mathcal{T}_{\text{required}}|} > 0.8$ consistently

- Sequence Phase: Convergence measured by sequence correctness score $> 0.7$ and step efficiency improvement

The parameter update incorporates gradient clipping ($\|\nabla_\theta J(\theta)\|_2 \leq 0.5$) and momentum-based optimization to ensure stable convergence across both training phases.

This policy update framework directly optimizes the multi-source confidence weighting to maximize the phase-adaptive reward signals, enabling the system to learn both tool discovery and sequence optimization through unified gradient-based learning.

### D.5.2 EXPERIENCE COLLECTION AND REPLAY

The system maintains episode trajectories for pattern learning:

$$\mathcal{D}_{\text{episode}} = \{(s_t, a_t, r_t, s_{t+1})_{t=0}^{T}\}$$
$$\mathcal{D}_{\text{patterns}} = \text{ExtractPatterns}(\mathcal{D}_{\text{episode}}, \text{score}_{\text{final}})$$
$$\mathcal{P}_{\text{successful}} = \mathcal{P}_{\text{successful}} \cup \mathcal{D}_{\text{patterns}}$$

**Pattern Extraction:** Sequential tool patterns are learned from successful episodes:

$$\text{pattern}_k = \text{sequence}[i : i + k] \quad \forall i, k \in \{2, 3\}$$
$$\text{score}(\text{pattern}_k) = \frac{\sum_{\text{episodes}} \text{score}_{\text{episode}} \cdot \mathbf{1}[\text{pattern}_k \in \text{episode}]}{\sum_{\text{episodes}} \mathbf{1}[\text{pattern}_k \in \text{episode}]}$$

### D.5.3 GENERATING THE MDP-OPTIMAL WORKFLOW PROMPT

Once an MDP-optimal workflow has been determined by the trained policy, it exists as a structured Python object containing a sequence of tools, dependency information, and rich metadata. To make this workflow actionable by a downstream execution agent, it must be translated into a comprehensive and unambiguous set of instructions. This is achieved through the `generate_mcp_prompt` function, which constructs a detailed prompt formatted according to a Multi-Agent Communication Protocol (MCP). The goal of this prompt is not merely to list the tools, but to provide a complete operational context, including the strategic reasoning, expected outcomes, and contingency plans. The generation process is composed of several automated steps that assemble distinct components of the final prompt.

**Core Structure** The entire prompt is encapsulated within a root `<mcp_task>` tag, creating a structured, machine-readable format. This structure is populated by several key sections, each generated by a dedicated helper function that extracts and formats information from the final workflow object.

**Execution Plan Generation** This is the central component of the prompt, outlining the sequence of actions the agent should take. The script employs an intelligent generation method, `_generate_smart_execution_plan`, to create a detailed, step-by-step guide. For each tool in the optimal sequence, this function populates the plan with a rich set of contextual details drawn from the `smart_actions` list in the workflow object:

- **Semantic Rationale:** If a tool was selected via semantic search (RAG), its relevance score is included (e.g., *Semantic match: 95%*). This immediately informs the agent of the confidence in that tool's applicability.

- **Generated Reasoning:** A concise, natural language explanation for the tool's selection is provided. This is generated by the `_generate_tool_reasoning` function, which considers factors like the tool's position in the sequence (e.g., *Initial data loading step*), its semantic capabilities (e.g., *Performs parse, transform operations*), and whether it was a mandatory requirement of the task.

- **Dependencies and Alternatives:** The prompt explicitly lists the direct prerequisites for each tool (e.g., *Requires: file_operations_reader*) and provides a list of alternative tools, ranked by semantic similarity, that can be used if the primary tool fails.

- **Expected Outcome and Confidence:** For each step, a high-level expected outcome is stated (e.g., *Expected: data_validated*), and a composite confidence score, calculated by `_calculate_comprehensive_confidence`, is displayed to manage the agent's expectations about potential failures.

**Intelligence and Semantic Insights** To provide the agent with meta-level awareness of the plan's quality, two sections are generated.

- The `<workflow_intelligence>` section provides a top-level summary, including the overall predicted success probability of the workflow, calculated by `_calculate_success_probability`.

- The `<semantic_insights>` section, generated by `_generate_semantic_insights`, offers a quantitative summary of how Retrieval-Augmented Generation (RAG) influenced the plan. It reports metrics such as the average semantic match score across all tools and the proportion of tools that were selected based on semantic relevance versus learned policy patterns.

**Contingency and Execution Guidance** The final sections of the prompt provide static but critical instructions to ensure robust execution.

- The `<failure_handling>` block provides a clear, universal protocol for error handling. It instructs the agent to first attempt the primary tool, then cycle through the provided alternatives upon failure, and finally report the issue if all options are exhausted.

- The `<execution_tracking>` block sets the requirements for the agent's response, instructing it to report its tool selection rationale, any deviations from the plan, and its confidence in each step's outcome. This ensures that the execution results can be used for future learning and analysis.

Through this multi-faceted generation process, a simple sequence of tools derived from an MDP is transformed into a rich, self-contained operational directive that guides not just the "what" but also the "why", "how", and "what if" of task execution.

# E DETAILED EXPERIMENTAL RESULTS

## E.1 DETAILED MODEL PERFORMANCE ACROSS TASK TYPE

The following table presents model performance across different task types. Performance generally decreases from basic file processing tasks to advanced computation pipelines, though patterns vary across model architectures.

Table 13: Model performance across task types.

| Task Type | GPT-4o-mini | GPT-5-mini | Claude-Sonnet-4 | Gemini-2.5-Flash | DeepSeek-V3 | DeepSeek-R1 | Qwen2.5-32B | Llama-3.3-70B |
|---|---|---|---|---|---|---|---|---|
| Advanced processing | 55.2% | 54.1% | 51.4% | 50.0% | 48.3% | 50.0% | 52.4% | 55.8% |
| API data retrieval | 59.0% | 54.6% | 50.0% | 57.1% | 49.5% | 47.5% | 57.1% | 64.3% |
| Batch processing | 82.0% | 58.7% | 60.5% | 64.6% | 58.6% | 50.0% | 64.0% | 59.8% |
| Content analysis | 77.9% | 76.3% | 66.0% | 70.9% | 70.0% | 66.1% | 82.1% | 81.7% |
| Multi-step processing | 67.5% | 62.1% | 51.7% | 59.5% | 59.5% | 47.3% | 72.9% | 65.3% |

## E.2 COMPLETE MODEL PERFORMANCE ON REAL-WORLD TEST SET

In this section, we present the complete model performance table, including results under the baseline prompt and CoT prompt.

Table 14: Prompt type performance comparison: Success rates across different prompt types for each model.

| Model | Baseline | | | Chain-of-Thought | | | Optimal Workflow | | | Flawed Workflow | | |
|---|---|---|---|---|---|---|---|---|---|---|---|---|
| | Full | Partial | Fail | Full | Partial | Fail | Full | Partial | Fail | Full | Partial | Fail |
| GPT-4o-mini | 38.9 | 38.9 | 22.2 | 41.7 | 41.7 | 16.7 | 42.1 | 38.6 | 19.3 | 34.3 | 34.3 | 31.4 |
| Gemini-2.5-Flash | 33.3 | 33.3 | 33.3 | 39.5 | 39.5 | 21.1 | 55.3 | 41.2 | 3.5 | 34.1 | 33.5 | 32.4 |
| GPT-5-mini | 38.9 | 38.9 | 22.2 | 41.7 | 41.7 | 16.7 | 47.6 | 45.2 | 7.1 | 36.6 | 36.3 | 27.0 |
| Llama-3.3-70B | 16.7 | 16.7 | 66.7 | 16.7 | 16.7 | 66.7 | 39.6 | 27.1 | 33.3 | 25.4 | 25.4 | 49.3 |
| Qwen2.5-32B | 41.7 | 41.7 | 16.7 | 43.9 | 40.4 | 15.8 | 55.3 | 44.7 | 0.0 | 42.3 | 38.8 | 18.9 |
| DeepSeek-V3 | 32.1 | 30.9 | 37.0 | 37.8 | 35.3 | 26.9 | 40.2 | 39.3 | 20.5 | 36.0 | 35.4 | 28.6 |
| Avg | 33.6 | 33.4 | 33.0 | 36.9 | 35.9 | 27.3 | **46.7** | 39.4 | 13.9 | 34.8 | 34.0 | 31.3 |

## E.3 SCALING ANALYSIS ON THE REAL WORLD TEST SET

In this section, we present scaling analysis results for the Qwen2.5 series on the real-world test set under MDP-optimal workflow prompts. Table 15 reveals a dramatic emergence of workflow execution capabilities between the 3B and 7B parameter scales. The 3B model exhibits near-complete failure (93.3% failure rate with only 5.0% full success), indicating insufficient capacity for multi-step API coordination. A sharp capability transition occurs at 7B parameters, where the model achieves 37.7% full success with failure rate dropping to 28.1%. The 14B model maintains similar performance (35.0% full success, 30.0% failure), while the 32B model demonstrates the strongest performance with 55.3% full success and zero failures. This non-smooth scaling pattern—characterized by sharp capability emergence between 3B and 7B, followed by steady improvement to 32B—provides evidence for emergent workflow execution abilities in real-world API interaction settings, corroborating our simulation findings (Section 4.4).

Table 15: Qwen2.5 series scaling analysis with Optimal prompt.

| Model Size | Full Success Rate | Partial Success Rate | Failure Rate |
|---|---|---|---|
| Qwen2.5-3B | 5.0% | 1.7% | 93.3% |
| Qwen2.5-7B | 37.7% | 34.2% | 28.1% |
| Qwen2.5-14B | 35.0% | 35.0% | 30.0% |
| Qwen2.5-32B | 55.3% | 44.7% | 0.0% |

## E.4 REAL-WORLD API FAILURE DISTRIBUTION

We present a record of 948 total API calls across 23 different live APIs, achieving an overall success rate of 52.7% (500 successful calls, 448 failures). This 47.3% natural failure rate demonstrates the inherent uncertainty in real-world API interactions that our benchmark aims to capture. Unlike the simulated tools in our main experiments, these APIs execute actual HTTP requests to live endpoints, exposing agents to real-world system behaviors. The error modes and system behaviors directly come from real-world API interactions. Table 16 shows the distribution of error types observed during our experiments.

Table 16: Error Type Distribution in API Calls

| Error Type | Counts | Percentage of Failures |
|---|---|---|
| TIMEOUT | 337 | 71.10% |
| OPERATION_FAILED | 41 | 8.65% |
| INVALID_INPUT | 27 | 5.70% |
| INVALID_RESPONSE | 21 | 4.43% |
| NETWORK_ERROR | 21 | 4.43% |
| RATE_LIMIT_ERROR | 1 | 0.21% |

We now present an explanation of all the error types presented in the table. The predominant error type, TIMEOUT, reflects our implementation of a 30-second timeout threshold for API responses.

This design choice aligns with common industry practices for production systems, where timeout mechanisms are essential safeguards against indefinite waiting and resource exhaustion. Importantly, the occurrence of timeout errors depends entirely on real-world network conditions, server load, and backend processing times, making these failures authentic reflections of the unpredictable nature of distributed systems rather than artificial constraints. `OPERATION_FAILED` errors result from Python exceptions during response processing, such as JSON parsing failures or encoding errors. `INVALID_INPUT` are mapped from HTTP 400 status codes returned by APIs when request parameters are missing or malformed. `INVALID_RESPONSE` errors occur when APIs return unparseable content, such as empty response bodies or data that doesn't match expected schemas. `NETWORK_ERROR` are triggered by issues including DNS resolution failures and connection refusals. `RATE_LIMIT_ERROR` directly map to HTTP 429 responses when API rate limits are exceeded.

## F  LIMITATIONS

Our simulation uses a simplified probabilistic failure model (base success rate 0.8 with penalty adjustments) that does not capture real-world complexities such as input-dependent errors, time-varying reliability, or realistic recovery mechanisms like exponential backoff. These simplifications balance reproducibility, scalability, and our focus on instruction quality variation. While the real-world task set (Section 3.4) shows simulation rankings correlate with real-world performance, future work could incorporate input-conditioned failure probabilities, explicit rate limiting, and correlated failure patterns.

## G  THE USE OF LARGE LANGUAGE MODELS

In adherence to the ICLR 2026 policy on the use of Large Language Models (LLMs), we disclose that an LLM was utilized as a specific component within our benchmark construction methodology.

**Role in Code Development.**   The implementation of our experimental framework, including the automated task generation pipeline and the simulation environment, was expedited with the assistance of LLM-based coding tools. Specifically, we utilized Anthropic's Claude for generating code snippets, debugging complex logic, and refactoring. These tools acted as programming assistants, and all final code was reviewed, validated, and integrated by the human authors.

**Role in Manuscript Preparation.**   Our writing process for the appendices involved a collaborative human-LLM workflow. The initial drafts of all appendix sections were written by the human authors to ensure the factual and technical accuracy of the content. These drafts were then processed by an LLM for the purpose of polishing the language, improving formal structure, and enhancing clarity. Following the LLM's revisions, the authors conducted a final, thorough review to make critical edits, verify all statements, and ensure the text precisely reflected our methodology and findings.

The core research ideation, experimental design, analysis of results, and the primary drafting of the main manuscript were conducted by the human authors. We take full and final responsibility for all content presented in this paper, including any code and text produced with the assistance of LLMs.

