# OpenReview forum: "ResiliBench: Evaluating Agentic Workflow Adaptation in Stochastic Environments"
_ICLR.cc/2026/Conference — ICLR 2026 Poster_

### Official Review · Reviewer_5aFN · 2025-10-27

**Soundness:** 3
**Presentation:** 2
**Contribution:** 3
**Rating:** 6
**Confidence:** 3

**Summary:**

This paper introduces a new benchmark PILOT-Bench, designed to evaluate the workflow execution capability of LLMs in tool-driven scenarios that more closely approximate real-world complexity. The authors argue that existing benchmarks primarily focus on idealized settings where tool behavior is deterministic and guidance is precise. To address this, PILOT-Bench introduces two core real-world challenges:

1.  Tool Execution Uncertainty: The benchmark simulates probabilistic tool behavior by parameterizing error models, reflecting failure modes that APIs may exhibit in the real world.
2.  Variability in Guidance Quality: The benchmark provides workflow instructions of varying quality, including "Optimal Workflow" prompts and "Defective Workflow" prompts, to assess the model's robustness to changes in instruction quality.

The benchmark comprises a tool library of 30 APIs, from which 5,040 tasks are generated. A significant contribution is that the Optimal Workflows are generated using a MDP framework that considers tool dependencies and stochastic behavior, with policy optimization performed using PPO. The Defective Workflows are created by systematically injecting seven categories of controlled perturbations into the optimal workflows.

**Strengths:**

1.  The paper points out the limitations of existing tool-use benchmarks. By focusing on probabilistic tool failures and variability in instruction quality, the benchmark addresses two core challenges for LLM agents in practical deployment.
2.  The experimental results are insightful. For instance, the high robustness exhibited by some state-of-the-art models when given defective prompts suggests that robustness to defective guidance might be an ability dimension independent of general tool-use capability. Furthermore, the negative impact of task complexity on performance aligns with expectations.

**Weaknesses:**

1.  One of the most confusing yet undiscussed results in the paper is that the "Complete Success Rate" of the Baseline (which only contains the task description) is actually higher than that of the Optimal Workflow for several state-of-the-art models. This directly challenges the premise that the MDP-generated workflow is "optimal." The authors do not seem to discuss this phenomenon. Does this imply that the level of detail and complexity in the optimal workflow actually interferes with or confuses the model, or is my understanding incorrect?
2.  While the introduction of probabilistic failure is a good mechanism, I believe the simulation is still relatively simple. The testing environment uses a base success rate of 0.8 with a penalty adjustment based on dependencies and historical failures. Real-world API failures are often not independent and can depend on factors like rate limits or specific errors caused by invalid inputs. The current model captures five failure types, but the simulation logic seems to simplify this into a single probability calculation.
3.  The "optimal" workflow is an artifact generated by an MDP agent that maximizes its own reward function within its training environment. It is unclear whether this strategy corresponds to the "gold standard" or the truly most efficient workflow as perceived by a human expert. Its optimality is relative to its training process.

**Questions:**

Please refer the weakness :-)

---

> ### Author Response · Authors · 2025-11-23
>
> ### Response to Weakness 1
>
> We thank the reviewer for highlighting this counter-intuitive result in our initial experiments. We have investigated this phenomenon and addressed it through both qualitative analysis and quantitative updates.
>
> **1. Behavioral Analysis of the “Caution Penalty”:** Our detailed trace analysis reveals that providing a granular execution plan (Optimal Workflow) induces a “verification-heavy” behavior in capable models. These models tend to verify the provided plan by performing excessive `tool_search` or `tool_info` queries before execution. In contrast, models given only the Baseline prompt often adopt a direct execution strategy. Under strict turn constraints, the cautious overhead of the MDP-Optimal Workflow occasionally led to incomplete tasks despite correct reasoning.
>
> **2. Updated Task Library and Results:** We hypothesized that the over-verification was partly triggered by ambiguities in the original task descriptions. We have since updated the task library to feature more concrete and comprehensive descriptions. As shown in the new experiments below (we have also updated this to Table 2 in the paper), the counter-intuitive result disappears. The *Optimal Workflow* now consistently outperforms the *Baseline* across state-of-the-art models (e.g., **GPT-4o-mini**: 67.7% vs 50.5%, **Gemini-2.5-Flash**: 60.1% vs 54.3%), confirming the efficacy of the MDP-generated plans.
>
> | **Model** | **Baseline Full** | **Baseline Partial** | **Baseline Fail** | **Optimal Full** | **Optimal Partial** | **Optimal Fail** |
> |-----------|-------------------|-----------------------|-------------------|-------------------|----------------------|------------------|
> | **GPT-4o-mini** | 50.5 | 46.5 | 3.0 | 67.7 | 31.2 | 1.1 |
> | **O3-0416-Global** | 52.7 | 44.9 | 2.4 | 58.5 | 35.1 | 6.4 |
> | **Gemini-2.5-Flash** | 54.3 | 44.5 | 1.1 | 60.1 | 36.7 | 3.3 |
> | **GPT-5-mini** | 52.0 | 46.0 | 2.0 | 60.7 | 35.5 | 3.8 |
> | **Llama-3.3-70B** | 47.8 | 42.5 | 9.6 | 66.1 | 30.9 | 3.0 |
> | **Qwen2.5-32B** | 52.5 | 43.8 | 3.7 | 65.0 | 31.9 | 3.1 |
> | **DeepSeek-V3** | 50.0 | 50.0 | 0.0 | 56.8 | 39.0 | 4.2 |
>
> ### Response to Weakness 2
>
> We appreciate this insightful observation regarding the realism of our failure simulation. We acknowledge that our initial simulation model employs simplified probability calculations, and we address this concern through multiple dimensions:
>
> **1. Real-World API Validation Addresses Core Concern**
>
> In our revised version, we introduce the **real-world task set** with 23 live public APIs from the `public-apis/public-apis` GitHub repository.
>
> In these real-world experiments, agents encounter *actual* failure patterns including:
> - **Rate limiting and HTTP errors**: APIs return authentic error responses (e.g., rate limits, server errors) requiring appropriate retry strategies
> - **Input-dependent failures**: Invalid parameters trigger specific validation errors rather than generic failures
> - **Non-stationary behavior**: API reliability varies with time-of-day, service load, and maintenance windows
>
> The real-world experiments provide a complementary evaluation perspective, capturing authentic deployment challenges and offering richer insights into model behavior across varied operational contexts.
>
> **2. Design Trade-offs: Simplicity vs. Realism**
>
> We acknowledge the simplification inherent in our base success rate of 0.8 with penalty adjustments. This design reflects deliberate trade-offs:
> - **Controllability**: Our parameterized model enables systematic ablation studies on how different failure rates affect model performance
> - **Scalability**: Simulating 5,040 tasks with complex inter-API dependencies and time-varying failure patterns would require orders of magnitude more computational resources
>
> The validation results suggest that capturing the existence and probability of failures—rather than their precise mechanistic origins—appears to be adequate in differentiating model capabilities in workflow execution under uncertainty.
>
> **3. Acknowledgment of our current limitations**
>
> Following the reviewer's feedback, we have added a dedicated **Limitations section (Appendix F)** to the revised manuscript that explicitly discusses the simplified nature of our probabilistic failure model compared to real-world API behaviors, and the trade-offs between simulation complexity and scalability that motivated these design choices.

---

> ### Author Response · Authors · 2025-11-23
>
> ### Response to Weakness 3
>
> The reviewer raises an important conceptual distinction. Technically, our workflows are optimal with respect to the MDP formulation—they maximize expected cumulative reward under our defined state space, action space, and reward function. However, the reviewer precisely points out that this is **MDP-optimality**, not absolute optimality. In this sense, our workflows optimize for:
> - Tool coverage (executing all required tools)
> - Sequence correctness (respecting dependencies)
> - Success rate in our simulated environment
>
> But not necessarily optimize for:
> - API cost or latency
> - Human-preferred execution patterns
> - Robustness to out-of-distribution scenarios
> - Other notions of “quality” not captured in our reward
>
> **Revised Terminology and Claims:**
>
> We have updated the manuscript to:
> - Replace "optimal workflow" in most contexts with "MDP-optimal workflow"
> - Clarify these workflows represent one instantiation of high-quality instructions, not ground truth
>
> We would like to additionally remark that the role of workflows in the benchmark serves to create a *controlled instruction quality spectrum*, with high-quality workflows derived from MDPs that are dependency-aware and achieve a high success rate, and flawed workflows that contain systematic errors. This setup enables systematically studying model robustness to instruction variation—our research question—without claiming that these workflows represent the “best possible” by all definitions.

---

### Official Review · Reviewer_eCNp · 2025-10-31

**Soundness:** 1
**Presentation:** 2
**Contribution:** 2
**Rating:** 2
**Confidence:** 4

**Summary:**

In this paper, the authors propose a benchmark named PILOT-Bench for evaluating large language models in tool-driven workflow scenarios under uncertainty. The benchmark comprises 5,040 tasks across 30 APIs, testing robustness to flawed instructions and stochastic tool behavior. Experiments across ten LLMs reveal success rates of 71.8% for optimal, 59.4% for Chain-of-Thought, and 61.5% for flawed workflows.

**Strengths:**

1. The paper introduces a comprehensive benchmark for evaluating LLM workflow execution under uncertain tool-use conditions, encompassing 5,040 tasks and 30 tool APIs.
2. The evaluation includes a wide range of models, covering both proprietary and open-source LLMs.

**Weaknesses:**

1. The result analysis lacks depth—although extensive data are presented (Sections 4.2–4.4), the discussion does not yield clear or insightful conclusions.
2. The claimed advantages over existing benchmarks (e.g., ToolBench) are limited, as prior works also incorporate probabilistic tool failures and instruction variability.

**Questions:**

Line 352 references the Qwen2.5 series, yet only Qwen2.5-32B results are reported.

---

> ### Author Response · Authors · 2025-11-23
>
> ### Response to Weakness 1
>
> We sincerely appreciate this valuable feedback. We have comprehensively reorganized the experiment section to provide deeper analytical insights and more meaningful conclusions from our experiment results. Our experimental analysis of LLM workflow execution capabilities demonstrates:
> - **Different robustness patterns to instruction quality**: Models exhibit dramatically different robustness patterns when confronted with flawed instructions. For example, GPT-4o-mini maintains relatively stable performance across instruction quality variations (optimal: 67.7%, flawed: 62.2%, a 5.5 percentage point drop), while Gemini-2.5-Flash shows substantial degradation (optimal: 60.1%, flawed: 20.0%, a 40.1 percentage point drop).
> - **Emergent abilities of workflow execution**: Through experiments on the Qwen2.5 series, we observe emergent abilities in workflow execution, where the model's multi-step workflow execution ability appears suddenly at certain parameter thresholds rather than scaling smoothly.
> - **Real-world validation**: We extend our evaluation with a real-world test set integrating live public APIs. Real-world experiments reveal patterns consistent with the observations from the simulated API experiments.
>
> We have also updated these summaries in the experiment section of our paper.
>
> ### Response to Weakness 2
>
> We thank the reviewer for this accurate observation. We acknowledge that existing benchmarks may encounter probabilistic tool failures and instruction variability as natural consequences. However, our work makes two distinct contributions:
>
> **1. Focus on Uncertainty as Primary Study Object:** Existing benchmarks typically filter and curate APIs or instructions to minimize tool failures and instruction variability [1][2][3]. In contrast, our approach makes these uncertainties the primary focus of systematic study. We deliberately introduce controlled perturbations and probabilistic error models to create environments that require adaptive strategies and error recovery. This shift in focus allows us to systematically evaluate how models handle uncertainty rather than simply measuring capability under ideal conditions.
>
> **2. Analytically-Derived Optimal Workflows via MDP:** We provide analytically derived optimal workflows through MDP optimization. Under realistic deployment constraints, there exist three theoretical upper bounds for success rates: (i) 100% may not be achievable due to round/retry limits; (ii) the upper bound achieved by a policy knowing tool call procedures and able to anticipate tool errors (e.g., knowing the random seed of every tool operation); (iii) the upper bound achieved by a policy knowing tool call procedures but only aware of the probability of tool failure. Our MDP formulation provides a tractable approximation to bound (iii), computing workflows that maximize expected success rates by reasoning over known reliability statistics. These MDP-optimized workflows serve as performance benchmarks under uncertainty.
>
> We have updated these statements in the introduction section and the related work section of the main text.
>
>
> ### Response to Question 1
>
> We sincerely appreciate this valuable feedback and apologize for the confusing expression. We have deleted the statement in Line 352 of the original paper, and we have now included a dedicated scaling analysis (Section 4.4) that comprehensively presents the complete Qwen2.5 series results across all model sizes (3B, 7B, 32B, and 72B) in Table 4 in the updated paper.
>
> ---
> [1] Li, Minghao, et al. "API-Bank: A comprehensive benchmark for tool-augmented LLMs." EMNLP 2023.
> [2] Huang, Yue, et al. "Metatool benchmark for large language models: Deciding whether to use tools and which to use." ICLR 2024.
> [3] Qin, Yujia, et al. "Toolllm: Facilitating large language models to master 16000+ real-world apis." ICLR 2024.

---

### Official Review · Reviewer_NTap · 2025-11-01

**Soundness:** 3
**Presentation:** 2
**Contribution:** 3
**Rating:** 4
**Confidence:** 3

**Summary:**

The paper introduces a benchmark PILOT-Bench for evaluating LLM workflow execution under probabilistic tool behaviors and variable instruction quality. PILOT-Bench contains 5,040 tasks generated from a library of 30 APIs with explicit probabilistic error modes such as TIMEOUT, DEPENDENCY ERROR, and INVALID INPUT. Experiments show that current LLMs performance differences when models encounter probabilistic tool failures and varying instruction quality.

**Strengths:**

1. The benchmark explicitly models stochastic tool behaviors and noisy instructions, which is novel from previous tool learning benchmarks. The probabilistic simulator and MDP-based workflow construction are well-motivated and technically sound.
2. The automated generation pipeline is clearly described. The inclusion of both optimal and flawed workflows enables the study of robustness to imperfect guidance, which is highly relevant for practical agents.
3. The experimental evaluation shows results of ten diverse LLMs which is solid and sound.

**Weaknesses:**

1. All experiments rely on a simulated environment. Demonstrations on real API tools or open-ended workflows would better substantiate the validity.
2. Writing has room to improve. Some sections  repeat definitions of tool errors and workflow generation, which is redundant.

**Questions:**

Could the benchmark be extended to adaptive settings where the LLM can retry or re-plan after a failure? Reference settings are like ToolEVO [1].

[1] Learning Evolving Tools for Large Language Models. ICLR 2025.

---

> ### Author Response · Authors · 2025-11-23
>
> ### Response to Weakness 1
>
> We appreciate the feedback regarding the necessity of validating our benchmark against real-world systems. To address this concern, we have extended PILOT-Bench with a **real-world
> task set** that provides complementary evaluation on live public APIs.
>
> **1. Integration of Live Public APIs**
>
> We integrated 23 functional APIs sourced from the `public-apis/public-apis` GitHub repository. Unlike the simulated tools in our main experiments, these APIs execute actual HTTP requests to live endpoints, exposing agents to real-world system behaviors. The error modes and system behaviors also directly come from real-world API interactions.
>
> **Example Mechanism**: Consider the `network_coinpaprika` tool. Instead of returning a pre-defined mock response, it connects directly to the Coinpaprika API endpoint. When an agent invokes this tool, it performs a real network call to fetch live cryptocurrency market data. This integration exposes the agent to genuine real-world dynamics, including actual network latency, live data structures and formats, and realistic HTTP status codes and API error responses.
>
> **2. Tasks and Experiment Results**
>
> We constructed 8 real-world sequential tasks and evaluated the representative models across both proprietary and open-source categories. As an example, the *content creation task* requires models to sequentially call four real-world APIs—fetching a random fact, a joke, a programming quote, and a stoic quote—then compile them into a social media post draft. We have updated the tasks and API-based tools to our GitHub repo: <https://github.com/PilotBenchAnonymous/PilotBench>.
>
> The results are presented in the table below (Real-world results), and we have updated the results in our paper. Real-world results align with our simulations, revealing that models exhibit different robustness patterns when confronted with flawed instructions.
>
> ### Real-world results
>
> | **Model** | **Optimal Workflow Full** | **Optimal Workflow Partial** | **Optimal Workflow Fail** | **Flawed Workflow Full** | **Flawed Workflow Partial** | **Flawed Workflow Fail** |
> |-----------|----------------------------|-------------------------------|----------------------------|----------------------------|-------------------------------|----------------------------|
> | **GPT-4o-mini** | 42.1 | 38.6 | 19.3 | 34.3 | 34.3 | 31.4 |
> | **Gemini-2.5-Flash** | 55.3 | 41.2 | 3.5 | 34.1 | 33.5 | 32.4 |
> | **GPT-5-mini** | 47.6 | 45.2 | 7.1 | 36.6 | 36.3 | 27.0 |
> | **Llama-3.3-70B** | 39.6 | 27.1 | 33.3 | 25.4 | 25.4 | 49.3 |
> | **Qwen2.5-32B** | 55.3 | 44.7 | 0.0 | 42.3 | 38.8 | 18.9 |
> | **DeepSeek-V3** | 40.2 | 39.3 | 20.5 | 36.0 | 35.4 | 28.6 |
> | **Avg** | 46.7 | 39.4 | 13.9 | 34.8 | 34.0 | 31.3 |
>
> ### Response to Weakness 2
>
> We thank the reviewer for this observation. We have
> addressed the redundancy by: (1) removing the duplicated error type
> enumeration from Section 3.1, and (2) simplifying the workflow description
> in Section 2.1 with a forward reference to the detailed methodology in
> Section 3.3.
>
> ### Response to Question 1
>
> We appreciate the question regarding the extension to a retry and re-plan setting. We would like to clarify that PILOT-Bench already incorporates adaptive settings that enable LLMs to retry and re-plan after failures, similar in spirit to ToolEVO's approach but with some key distinctions.
>
> Our framework provides several adaptive mechanisms:
> 1. **Multi-turn interactive execution**: Models can interact for up to 10 conversational turns per task, allowing multiple attempts and strategy adjustments based on environmental feedback.
> 2. **Real-time error feedback and recovery**: When tools fail (e.g., due to `TIMEOUT`, `DEPENDENCY_ERROR`, or `INVALID_INPUT`), the system provides detailed error messages and recommendations, enabling models to adapt their approach. As demonstrated in our execution examples, when a model encounters dependency errors, it can search for alternative tools or retry with corrected parameters.
> 3. **Dynamic tool discovery**: Models can use `<tool_search>query</tool_search>` at any turn to discover new tools or re-explore available options when their initial plan fails.
>
> Importantly, our workflow prompts serve as **instructions rather than rigid requirements**, instructing models: *"1. Analyze the task requirements
> 2. Search for appropriate tools based on what you need to do
> 3. Execute the tools in the correct order
> 4. Complete the task and indicate when finished"* and *"Use alternatives if primary tools fail"*. The text in *italics* represents the exact wording from the prompts. This design philosophy aligns with ToolEVO's on adaptive behavior.

---

> > ### Comment · Reviewer_NTap · 2025-11-24
> >
> > I appreciate that the authors added the real-world API experiments. However, it would be helpful to report basic statistics of the live API calls, such as overall failure rate and the distribution of error types.

---

> ### Author Response · Authors · 2025-12-02
>
> We thank the reviewer for this excellent suggestion. We now present a record of 974 total API calls across 23 different live APIs, achieving an overall success rate of 51.3% (500 successful calls, 474 failures). This 48.7% natural failure rate demonstrates the inherent uncertainty in real-world API interactions that our benchmark aims to capture.
> Unlike the simulated tools in our main experiments, these APIs execute actual HTTP requests to live endpoints, exposing agents to real-world system behaviors. The error modes and system behaviors directly come from real-world API interactions. The table below (Error Type Distribution in API Calls) shows the distribution of error types observed during our experiments.
>
> ### Error Type Distribution in API Calls
>
> | **Error Type** | **Counts** | **Percentage of Failures** |
> |----------------|------------|-----------------------------|
> | `TIMEOUT` | 337 | 71.10% |
> | `OPERATION_FAILED` | 41 | 8.65% |
> | `INVALID_INPUT` | 27 | 5.70% |
> | `INVALID_RESPONSE` | 21 | 4.43% |
> | `NETWORK_ERROR` | 21 | 4.43% |
> | `RATE_LIMIT_ERROR` | 1 | 0.21% |
>
> We now present an explanation of all the error types presented in the table above. The predominant error type, `TIMEOUT`, reflects our implementation of a 30-second timeout threshold for API responses. This design choice aligns with common industry practices for production systems, where timeout mechanisms are essential safeguards against indefinite waiting and resource exhaustion. Importantly, the occurrence of timeout errors depends entirely on real-world network conditions, server load, and backend processing times, making these failures authentic reflections of the unpredictable nature of distributed systems rather than artificial constraints. `OPERATION_FAILED` errors result from Python exceptions during response processing, such as JSON parsing failures or encoding errors. `INVALID_INPUT` errors are mapped from HTTP 400 status codes returned by APIs when request parameters are missing or malformed. `INVALID_RESPONSE` errors occur when APIs return unparseable content, such as empty response bodies or data that doesn't match expected schemas. `NETWORK_ERROR` errors are triggered by issues including DNS resolution failures and connection refusals. `RATE_LIMIT_ERROR` errors directly map to HTTP 429 responses when API rate limits are exceeded.
>
> We have also added the API failure data to Appendix Section E.4 of our paper.

---

### Author Response · Authors · 2025-12-03
**Summary for Area Chair**

Following the recent ICLR security incident and subsequent paper reassignment, we sincerely thank you for the tremendous time and effort you spent on the submissions. To make your assessment easier, we have included a concise summary of the paper, reviews, our rebuttal, as well as the paper's revision changes.



## Paper Overview and Contributions
We introduce PILOT-Bench to evaluate LLM workflow execution under **realistic conditions**, especially **tool uncertainties and variations in instruction quality.** We achieved this by probabilistically modeling API errors and a systematic hierarchy of instruction qualities.

Prior works minimized such real-world noise [1][2][3]. ToolBench [3] performed a "rigorous filtering process" to ensure that its tool set "is reliable and functional," filtering from 53,190 APIs down to 16,464 high-quality APIs. We, however, don't dodge real-world challenges. We **systematically study these to measure the model's ability to recover from API errors and adaptively replan the workflow to complete tasks**.

Our benchmark encompasses 5,040 tasks derived from 30 APIs with integrated probabilistic failure patterns, as well as MDP-optimized workflows and controlled perturbation variants.



### Key Contributions
1. **Benchmark Design**: PILOT-Bench evaluates LLM workflow execution capabilities under instruction variability and tool uncertainty. Tasks are automatically generated.
2. **MDP-Based Workflow Generation**: We develop a Markov Decision Process (MDP) framework that generates **theoretically optimal execution workflows maximizing expected success rates**. By applying perturbations to the optimal workflow to systematically degrade the instruction quality, we evaluate LLM's robustness and tolerance to instruction imperfections.
3. **Evaluation Findings**: Models show different robustness to flawed instructions. GPT-4o-mini maintains stable performance (optimal: 67.7%, flawed: 62.2%); Gemini-2.5-Flash experiences severe degradation (optimal: 60.1%, flawed: 20.0%). In Qwen models, emergent workflow execution abilities appear suddenly at certain parameter thresholds.
4. **Real-World API Integration**: We revised PILOT-Bench to include 23 live public APIs that execute actual HTTP requests. Real-world experiments show patterns that are consistent with those observed in the simulated API experiments.

---

> ### Author Response · Authors · 2025-12-03
> **Summary of Response to Reviewer Feedback**
>
> ### Reviewer NTap
> **Concerns**:
>
> + Complement simulated tools with real-world APIs
> + Refine presentations to reduce redundancies
> + Extend adaptive settings with retry and re-planning capabilities
>
> **Response**:
>
> + **Real-World APIs**: We added 23 live public APIs executing actual HTTP requests. We recorded 974 calls with 51.3% success rate, observing natural failures including timeouts, operation failures, and network errors. Real-world experiments show patterns consistent with the observations from the simulated API experiments.
> + **Presentation**: We have removed redundant content.
> + **Adaptive Settings**: We clarify that PILOT-Bench already includes: (1) multi-turn execution and (2) tool error feedback for retry and re-planning. Workflow prompts serve as instructions, **not** rigid requirements, allowing adaptive strategies.
>
> ---
> ### Reviewer eCNp
> **Concerns**:
>
> + Demand deeper insights from experiment results
> + Request for clearer positioning relative to existing benchmarks
> + Inquiry about complete Qwen 2.5 series results
>
> **Response:**
>
> **Enhanced Analysis**: We added more insights derived from our evaluation. Three key findings are:
>
> 1. **Different robustness to instruction quality**: Models exhibit dramatically different robustness patterns when confronted with flawed instructions. For example, GPT-4o-mini maintains relatively stable performance (67.7% to 62.2%), while Gemini-2.5-Flash experiences much more degradation (60.1% to 20.0%).
> 2. **Emergent abilities in workflow execution**: Through systematic evaluation of the Qwen 2.5 series across different model sizes, we observe multi-step workflow execution proficiency appears suddenly at certain parameter thresholds rather than improving gradually with model scale.
> 3. **Real-world validation**: As mentioned, we extended our evaluation with 23 live public APIs executing actual HTTP requests. Real-world experiments are consistent with simulated API experiments.
>
> + **Clearer Positioning**: We clarify that PILOT-Bench fills a critical practical gap by explicitly modeling the realistic conditions—specifically tool reliability and instruction variability—that prior benchmarks [1][2][3] aimed to eliminate.
>
> Additionally, we establish a theoretical performance ceiling through MDP optimization. Instead of assuming an unrealistic 100% success rate or omniscient error prediction, our framework calculates the maximum achievable success rate given the tools' probabilistic failure modes. These analytically derived workflows serve as a grounded upper bound for evaluation.
>
> + **Complete Results**: We moved the comprehensive Qwen2.5 scaling analysis (3B, 7B, 32B, 72B) to the main text (Section 4.4, Table 4).
>
> ---
> ### Reviewer 5aFN
> **Concerns**:
>
> + Counter-intuitive results of baseline prompt outperforming MDP-optimal workflow prompt
> + Tool failure simulation oversimplifies real-world API complexity into probability calculations.
> + Clarification is needed regarding the scope of MDP-generated workflow optimality
>
> **Response**:
>
> + **First concern—explanation and fix**: We observed that optimal workflow prompts, which are more detailed than baseline prompts, triggered excessive verification, running into turn limits. After updating our task library with clearer task descriptions, we now observe that optimal workflows consistently outperform baselines.
> + **Explaining our simplification rationale**: While causes of failures are countless, the end result is a probability of failure. After our revision incorporated real-world APIs with authentic, complex failure patterns, we observed patterns in the real-world experiments that are consistent with those from the simulated API experiments.
> + **Optimality Scope**: We now use "MDP-optimal" throughout, clarifying that these workflows maximize expected reward within our MDP formulation. They optimize for tool coverage, dependencies, and success rates, but not all possible metrics (cost, latency, robustness to novel scenarios).

---

> ### Author Response · Authors · 2025-12-03
> **Key Takeaways**
>
> PILOT-Bench fills a critical practical gap by evaluating LLM workflow execution under realistic conditions, especially **tool uncertainties** and **variations in instruction quality**. By combining a **theoretically grounded MDP framework** with our newly integrated **live API experiments**, we offer a robust standard for measuring true agent reliability. We have diligently addressed all reviewer concerns— expanding the scope of our experiments and clarifying our theoretical contributions. Given the benchmark’s potential to guide the development of resilient, real-world-ready agents, we sincerely hope that the Area Chair will consider our work favorably.
>
> ---
> [1] Li et al. "API-Bank: A comprehensive benchmark for tool-augmented LLMs." EMNLP 2023.
> [2] Huang et al. "Metatool benchmark for large language models." ICLR 2024.
> [3] Qin et al. "Toolllm: Facilitating large language models to master 16000+ real-world apis." ICLR 2024.

---

### Meta-Review · Area_Chair_M9EU · 2025-12-29

**Summary:**

The paper introduces PILOT-Bench, a benchmark designed to evaluate LLM workflow execution under realistic conditions characterized by tool execution uncertainty and variable instruction quality. Distinct from prior benchmarks that typically filter out noise to ensure stability, this work explicitly models probabilistic tool behaviors and systematically injects instruction imperfections. The authors propose a Markov Decision Process framework to generate theoretically optimal workflows that maximize expected success rates, providing a grounded upper bound for evaluation. Extensive experiments across various LLMs reveal significant disparities in model robustness to flawed instructions and highlight emergent capabilities in complex workflow execution.

**Reviewer Concerns:**

1. A primary concern raised by Reviewers NTap and 5aFN was the reliance on a simulated environment. They questioned whether the probabilistic error models could adequately capture the complexity of real-world API failures, such as non-independent errors or rate limits, and requested validation against actual systems. The authors responded with a substantial update, integrating 23 live public APIs that execute actual HTTP requests and constructing a new real-world task set. They provided detailed statistics showing a natural failure rate of ~49% with diverse error types (e.g., timeouts, network errors) that mirrored the patterns observed in their simulation. From a reviewer's perspective, this empirical evidence is highly convincing; it bridges the gap between simulation and reality, effectively resolving the concern regarding the validity of the environment.

2. The novelty and positioning of the benchmark were challenged by Reviewer eCNp, who noted that existing benchmarks like ToolBench also encounter probabilistic failures. The reviewer felt the contribution was limited if it merely replicated existing challenges. In response, the authors clarified a fundamental methodological distinction: while prior works often view uncertainty as noise to be minimized or filtered, PILOT-Bench treats uncertainty as the primary object of study. They further differentiated their work by providing analytically derived performance upper bounds via the MDP framework, a theoretical contribution absent in previous benchmarks. While "novelty" is subjective, the authors' argument is strong. They successfully framed their contribution not just as "another dataset," but as a new evaluation paradigm (robustness under uncertainty) supported by a theoretical framework. This effectively addresses the critique by highlighting a unique value proposition that existing benchmarks do not offer.

3. Reviewer 5aFN identified a critical technical anomaly where the baseline prompts surprisingly outperformed the "optimal" workflows for several SOTA models, which undermined the validity of the MDP framework. Simultaneously, Reviewer eCNp criticized the analysis for lacking depth. The authors investigated and found that the original "optimal" prompts were overly detailed, causing capable models to engage in excessive verification that hit turn limits. By updating the task library with clearer descriptions, they demonstrated that the optimal workflows now consistently outperform baselines, thereby fixing the technical flaw. Additionally, they rewrote the analysis to highlight specific insights, such as emergent capabilities in the Qwen2.5 series (providing the missing scaling results requested) and distinct robustness patterns across models. These changes transform the experimental section from a simple data dump into a source of validated, insightful conclusions.

4. Finally, minor issues regarding presentation and terminology were raised, such as redundancy in text (Reviewer NTap) and the definition of "optimality" (Reviewer 5aFN). The authors streamlined the paper and adopted the precise term "MDP-optimal" to clarify that the workflows maximize reward within the mathematical formulation rather than representing a human "gold standard." These revisions show attention to detail and improve the paper's clarity and rigor.

**Reviewer Scores:**

* **Reviewer NTap: 4 to 6**. The reviewer's main condition for acceptance was real-world validation. The authors provided a robust set of real API experiments and detailed failure statistics, directly satisfying this request.


* **Reviewer eCNp: 2 to 4 or 6**. The authors addressed the "lack of depth" by rewriting the analysis and adding missing data. Regarding novelty, the clarification of "studying uncertainty" vs. "filtering noise" combined with the theoretical MDP contribution is persuasive. While the reviewer started with a strong rejection, the objective improvements to the paper's substance address the issue to some extent.


* **Reviewer 5aFN: 6 to 6**. This reviewer was already leaning positive but was held back by the counter-intuitive experimental result (Baseline > Optimal).

---

### Decision · Program_Chairs · 2026-01-26

Accept (Poster)